# Physicochemical, Rheological, and Nutritional Quality of Artisanal Fermented Milk Beverages with Cupuassu (*Theobroma grandiflorum*) Pulp and Flour

**DOI:** 10.3390/foods12112217

**Published:** 2023-05-31

**Authors:** Katherine Gutiérrez-Álzate, Iuri L. S. Rosario, Rafael L. C. de Jesus, Leonardo F. Maciel, Stefanie A. Santos, Carolina O. de Souza, Carla Paulo Vieira, Carlos P. Cavalheiro, Marion Pereira da Costa

**Affiliations:** 1Program in Food Science (PGAli), Faculty of Pharmacy, Federal University of Bahia (UFBA), Salvador CEP 40170-115, BA, Brazil; katherinega@ufba.br (K.G.-Á.);; 2Laboratorio de Inspeção e Tecnologia de Leite e Derivados (LAITLACTEOS), Federal University of Bahia (UFBA), Salvador CEP 40170-110, BA, Brazil; 3Laboratory of Cardiovascular Physiology and Pharmacology, Federal University of Bahia, Salvador 40110-902, BA, Brazil; 4Faculty of Pharmacy, Federal University of Bahia, Barão de Jeremoabo, 147, Ondina, Salvador CEP 40170-115, BA, Brazil; 5Department of Animal Science, Federal University of Bahia, Av. Adhemar de Barros, 500, Ondina, Salvador CEP 40170-110, BA, Brazil; 6Laboratory of Advanced Analysis in Biochemistry and Molecular Biology (LAABBM), Department of Biochemistry, Federal University of Rio de Janeiro (UFRJ), Cidade Universitária, Rio de Janeiro CEP 21941-909, RJ, Brazil; 7Laboratório de Inspeção e Tecnologia de Carnes e Derivados (LabCarne), Federal University of Bahia (UFBA), Salvador CEP 40170-110, BA, Brazil

**Keywords:** cupuassu, *Theobroma grandiflorum*, milk beverage, by-products, color analysis, rheological behavior, sensory acceptance

## Abstract

The use of fruits and their by-products in food has dramatically impacted the food industry due to the nutritional benefits and the technological and sensory effects of food matrices. Therefore, this research aimed to evaluate the effects of adding cupuassu (*Theobroma grandiflorum*) pulp and flour on fermented milk beverages’ physicochemical, microbial, and sensory properties during refrigerated storage (0, 7, 14, 21, and 28 days). Twelve formulations were realized with different percentages of cupuassu pulp (0, 5, 7.5, and 10% *w*/*v*) and flour (0, 1.5, and 3% *w*/*v*). The treatments with 3% cupuassu flour presented the highest percentages of protein, fat, fiber, and carbohydrates, compared with the samples containing pulp. On the other hand, the addition of pulp increased water retention capacity and color parameters (*L**, *a**, *b**, and *C**) and decreased pH and syneresis on day 0 of storage. During storage, the samples with pulp showed increases in pH values, consistency index, and apparent viscosity. In comparison, cupuassu flour addition decreased syneresis values and increased *L** and *b** during storage, as did pulp. In addition, sample HPHF (10% pulp and 3% cupuassu flour), based on just-about-right, penalty, and check-all-that-apply analyses, improved some sensory attributes of the fermented milk beverage, such as brown color, acid taste, bitter taste, cupuassu flavor, and firm texture. It can be concluded that cupuassu pulp and flour addition improves the physicochemical and sensory quality of fermented milk beverages and can provide nutritional value to the product.

## 1. Introduction

Artisanal fermented milk constitutes a diverse category of foods created by introducing fermenting bacterial or yeast stains into the milk of different species, resulting in products with different sensory properties, comprising acidic, sour, and tangy flavors and fluid, viscous, and thick textures [1]. Some of the most popular artisanal fermented milk products include fermented milk beverages, yogurt, kefir, koumiss, and buttermilk [2]. Additionally, many artisanal fermented milk products can be made using traditional methods and locally sourced ingredients, making them a sustainable and eco-friendly choice for consumers who are also looking to support small-scale farmers and producers. For instance, fermented milk beverages are a well-consumed group of dairy foods in Brazil that must contain a minimum milk base concentration of 51% *v*/*v* [3]. In this context, the addition of cheese whey, a by-product of the cheese industry that is generally discarded, as an ingredient in artisanal fermented milk beverages and other healthy food products due to its nutritional quality can be an alternative to reduce waste. Whey retains 50–55% of milk solids; 70–72% of lactose; 12–15% of water-soluble vitamins (B and C) and minerals (calcium, potassium, sodium, and magnesium); and 8–10% of proteins (α-lactalbumin, β-lactoglobulin, serum bovine albumin, glycomacropeptide, immunoglobulins, lactoperoxidase, and lactoferrin), protein-derived peptides (β-Lactophorin, β-Lactotensin, α-Lactophorin, Albutensin Serophorin, and Lactoferricin) and lipids, which provide innumerable health benefits [4,5,6,7]. Therefore, whey has the potential to be used as a raw material in developing new artisanal milk beverages that are not only distinctive in flavor and texture but also sustainable and health-promoting.

Moreover, fruit preparations, starches, pectins, xylooligosaccharides, and flours are reported to improve milk beverages’ sensory acceptability and physicochemical and rheological properties [8,9,10]. For instance, cupuassu (*Theobroma grandiflorum*) has been used in recent years for this purpose [11,12,13]. Its pulp has acidic flavor and intense fragrance that give distinctive flavor compared with other fruits that are more commonly used for yogurt production, such as strawberry and cantaloupe [14,15,16]. Likewise, cupuassu can improve some physicochemical properties of milk beverages, such as viscosity and water retention capacity (by decreasing syneresis) [11,13] due to its carbohydrate content (starch, pectin, and polysaccharides, among others), which gives high nutritional value to cupuassu pulp, as well as fatty acids (palmitic, linoleic, and α-linolenic acids), ascorbic acid (96–111 mg/100 g), and phenolic compounds (20.5 mg/100 g). Moreover, it has a bio-functional potential due to soluble fiber content [15,17] and antioxidant capacity (1.7–2.0 μM Trolox/g) [18]. Therefore, cupuassu can add nutritional and bio-functional value to dairy products.

Cupuassu flour is a by-product of the fruit obtained from the fermentation, roasting, and grinding of seeds (up to 17% of the fruit) [18]. This flour mainly comprises proteins (14.2–17.2%), lipids (28.2–36.3%), minerals (3.80%), and fibers (22.2%). Additionally, it has phenolic compounds (16.9 mg_GAE_/g_DM_), flavonoids (5.92 mg_CA_/g_DM_), epicatechin (20.74 mg/100 g_DM_), and antioxidant activity in ABTS and DPPH (151 mg_TEAC_/100 g_CE_ and 85.4 mM Trolox eq/L, respectively) [12,19]. Thus, cupuassu flour can be an interesting by-product to add value to fermented milk beverages.

The use of cupuassu pulp in different food products has been employed to improve the sensory properties of some dairy products, such as goat milk yogurt [11,20], and probiotic beverages [21], which are associated with various health benefits that are mainly attributed to phenolic compounds, such as flavones, flavonols, catechins, epicatechin, and protoanthocyanidin, which favor the consumer’s well-being with antioxidant activity and by reducing oxidative damage to lymphocyte DNA, presenting anticancer, antimicrobial, anti-inflammatory, and digestive stimulation effects [22,23] and allowing cardiovascular and circulatory diseases, cancer, diabetes, and Alzheimer’s and Parkinson’s diseases to be prevented [24,25,26].

Besides their potential nutritional, sensory, and techno-functional benefits, cupuassu pulp and flour are not widely used in developing new products. The primary hypothesis of this work is that cupuassu pulp and flour addition enhances the nutritional, sensory, and techno-functional quality of fermented milk beverages. Therefore, this work aimed to evaluate the effect of cupuassu pulp and flour on the physicochemical, microbial, and sensory properties of a fermented milk beverage prepared with milk and whey during 28 days at 4 °C.

## 2. Materials and Methods

### 2.1. Materials

Raw cow milk was obtained from the Experimental Farm of Entre Rios of Federal University of Bahia (UFBA). The sweet whey (pH 6.29 ± 0.02) obtained from the processing of fresh cheese and used in this study had the following proximal composition (Appendix A): dry matter (7.19 ± 0.07), lactose (3.96 ± 0.04), protein (2.65 ± 0.02), solids (0.58 ± 0.01), and fat (0.23 ± 0.01) in percent, with a density of 25.7 ± 0.02 kg/m^3^ and a freezing point of −0.44 ± 0.02 °C. The cupuassu pulp used was 100% natural, containing no water, sugar, nor preservatives, and was purchased at the Oliva Distribuidora store in Salvador de Bahia. The flour was packaged and distributed by C2 Alimentos and was purchased at local markets in São Paulo (Brazil).

### 2.2. Preparation of Artisanal Fermented Milk Beverages

Artisanal fermented milk beverages were prepared as described by Costa et al. [11] with modifications. In all treatments, pasteurized whey (49%) and milk (51%) were used as the liquid base, and sugar (3%) and lactic culture (0.1%; Fegurte 3^®^, Fego Alimentos LTDA, Goiânia, Brazil) composed of *Streptococcus thermophilus* and *Lactobacillus delbrueckii* subsp. *bulgaricus* were added. The mixture of ingredients was fermented in a water bath (model N1030; Centauro, Atuba, Pinhais, PR, Brazil) at a constant temperature (42 ± 1 °C for 5 h) until it reached a pH of 4.6. After interrupting the fermentation process, cupuassu pulp (water 30% and pulp 70%) and/or flour were added to the fermented milk beverages at different concentrations according to Costa et al. [20] and Jovanović et al. [27], depending on the treatment. Therefore, twelve treatments were performed: (1) beverage without added pulp and flour was considered the control (NPNF); (2) beverage with 1.5% flour (NPLF); (3) beverage with 3% flour (NPHF); (4) beverage with 5% pulp (LPNF); (5) beverage with 7.5% pulp (MPNF); (6) beverage with 10% pulp (HPNF). In addition, with a low concentration of flour combined with low, medium, and high concentrations of pulp, the treatments were as follows: (7) beverage with 10% pulp and 1.5% flour (HPLF); (8) beverage with 7.5% pulp and 1.5% flour (MPLF); (9) beverage with 5% pulp and 1.5% flour (LPLF). Finally, with a high concentration of flour combined with low, medium, and high concentrations of pulp, the following treatments were performed: (10) beverage with 10% pulp and 3% flour (HPHF); (11) beverage with 7.5% pulp and 3% flour (MPHF); (12) beverage with 5% pulp and 3% flour (LPHF). Lastly, the products were stored at 4 ± 1 °C for 28 days.

### 2.3. Proximate Composition

The proximate composition of the fermented milk beverages was analyzed on day 0. Moisture content was evaluated using the oven drying method; ash content, by ashing at 550 °C; protein content, with the Kjeldahl method, using 6.38 as a factor; and fat content, using soxhlet AOAC [28], where the results were expressed in %. Neutral Detergent Fiber (NDF) was analyzed in 0.500 g of sample with a EQ LCC 08 fiber analyzer (Tecnal, Piracicaba, Brazil) using the Ankom system with filter bags [29]. Carbohydrate content was determined by difference, subtracting the sum of the values previously obtained from 100% [27].

### 2.4. pH, Syneresis, Water-Holding Capacity (WHC), and Instrumental Color

pH, syneresis, water-holding capacity, and instrumental color were evaluated every seven days during storage (0, 7, 14, 21, and 28).

pH was determined with a bench pH meter for aqueous solutions (Model Mpa-210; Tecnopon, Piracicaba, São Paulo, Brazil). Before use, the electrode was calibrated with buffer solutions with pH 4.0, 7.0, and 10.0.

Syneresis and water-holding capacity were evaluated using an Eppendorf laboratory centrifuge, model 5702R (Eppendorf Ltd., Stevenage, UK), according to Ladjvardi [30] with slight modifications. Aliquots of 5 g of fermented milk beverages were centrifuged at 3000× *g* for 30 min at 10 °C, and the whey was separated. Syneresis (%) was calculated as the weight of generated supernatant (whey) per weight of fermented milk beverage multiplied by 100. In contrast, WHC was given as the percent weight of drained gel (precipitate) relative to the original weight of fermented milk beverages [31].

The values of lightness (*L**; 100 = white and 0 = black), redness (*a**; + red and - green), and yellowness (*b**; + yellow and - blue) of the fermented milk beverages were recorded with a Chroma Meter CR-5 colorimeter (Konica Minolta Business Technologies Inc., Tokyo, Japan) according to Costa et al. [11]. A D-65 light source, a 10° standard observer, and a 26 mm measuring area with a rectangular optical glass cell (lengths of 2 mm, 10 mm, and 20 mm) were employed for the precise measurement of liquid transmittance.

Chroma (*C**), hue angle (*h°*), and total color difference (Δ*E**) were calculated based on the analyzed color coordinates (Equations (1)–(3)). In addition, Δ*E* was calculated by matching the spectrum of the freshly prepared fermented milk beverage (day 0) and its relative spectrum on subsequent storage days according to Lucas et al. [32] as follows:(1)$$h{}^{\circ}=\arctan\left(\frac{b^{*}}{a^{*}}\right)$$
(2)$$C^{*}=\sqrt{\left({a^{*}}^{2}+{b^{*}}^{2}\right)}$$
(3)$$\Delta E=\sqrt{{\left(L_{n\;}^{*}-L_{0}^{*}\right)}^{2\;}+{\left(a_{n}^{*}-a_{0}^{*}\right)}^{2\;}+{\left(b_{n}^{*}-b_{0}^{*}\right)}^{2}}$$

### 2.5. Rheological Behavior

Rheological measurements were performed using a rheometer (Haake Rheotest; Mod. 2.1; Medingen, Germany) with concentric cylinders coupled to a water bath for temperature control (25 °C). The flow curves of samples were determined at speeds between 0.56 and 243 rpm; their corresponding shear rates (γ) and shear stresses (σ) were computed from relations given by the instrument’s manufacturer and then recorded. The experimental data were fitted to the Ostwald–de Waele model [33] as in Equation (4).
σ = Kγ^(n)^,(4)
where σ is the shear stress (Pa), K is the consistency index (Pa·s^n^), γ is the shear rate (s^−1^), and n is the flow behavior index (dimensionless). This model was used to evaluate the pseudoplastic behavior of each dairy beverage sample.

The rheological data were adapted to the Ostwald–de Waele model [34] to obtain viscosity values, which were reported as the apparent viscosity of samples, in the upward viscosity/shear rate curves at a shear rate between 25 and 1000 s^−1^, as in Equation (5).
μ = Kγ^(n−1)^,(5)
where μ is the apparent viscosity, K is the consistency index, γ is the shear rate, and n is the flow behavior index. The viscosity results were expressed in mPa.

### 2.6. Microbiological Analysis

All samples were analyzed for coagulase-positive *Staphylococcus*, molds and yeasts, *Salmonella* spp., total and thermotolerant coliforms, and *Escherichia coli* on days 1 and 28 of storage according to the American Public Health Association [35]. Briefly, 10 mL of each fermented milk beverage was homogenized in 90 mL of 0.1% peptone water. The samples were then subjected to serial dilutions and inoculated on Petri dishes using a Drigalsky loop. The enumeration of coagulase-positive *Staphylococcus* was performed on Baird Parker agar (Neogen, Lansing, MI, USA) after incubation at 35–37 ± 1 °C for 48 h. Mold and yeast counts were determined according to growth on Dichloran Rose Bengal Chloramphenicol Base (DRBC) agar and incubated aerobically at 25 ± 1 °C for 3–5 days. The colonies were expressed as log colony-forming units (CFU) per mL. For the analysis of *Salmonella* spp., samples were pre-enriched with 1% buffered peptone water (Merck KGaA, Darmstadt, Germany) and incubated at 35 ± 2 °C for 24 h, followed by selective enrichment in Tetrathionate broth (Neogen, Lansing, MI, USA) and Rappaport Vassiliadis broth (RV-Micromed Isofar) (Merck KGaA, Darmstadt, Germany) at 42 ± 2 °C for 24 h. Afterwards, aliquots were streaked on Xylose Lysine Deoxycholate agar (Neogen, Lansing, MI, USA), Hektoen enteric agar (Ionlab, Araucária, PR, Brazil), and Bright Green agar (Neogen, Lansing, MI, USA) and incubated at 35 ± 2 °C for 24 h.

Finally, the analyses of total and thermotolerant coliforms and *Escherichia coli* were performed following the most probable number (NMP) technique with Lauryl Sulfate Tryptose (LST) broth (Neogen, Lansing, MI, USA), followed by streaking on Eosine Methylene Blue Agar (EMB) for *E. coli* identification.

### 2.7. Sensory Analysis

For the sensory analysis, six treatments (HPHF, MPHF, LPHF, HPLF, MPLF, and LPLF) were used. These artisanal treatments were selected aiming to measure the sensory effect generated by cupuassu flour in combination with pulp on fermented beverages, based on previous studies reported by Costa et al. [11,20], who evaluated the same pulp concentrations, where its acceptability in fermented dairy products was evidenced.

Sensory analysis was conducted by the ethical norms for research on humans (National Health Council, Resolution No. 196/1996) and after approval by the Ethics Committee of School of Nursing of Federal University of Bahia (CAAE 60414022.7.0000.5531; approved on 1 March 2023). The adherence of individuals to work was ensured with the signing of the Term of Free and Informed Consent form. The sensory panel consisted of 61 untrained participants (38 women and 23 men), aged 18 to 56 years, who were regular consumers of dairy products. Persons with allergy or lactose intolerance were not recruited. All participants signed the informed consent form. All participants were recruited from Escola de Medicina Veterinária e Zootecnia (Universidade Federal da Bahia, Brazil). They evaluated the product with a quantitative acceptance test, using a 9-point hedonic scale, where 1 meant “I dislike it very much” and 9 meant “I like it very much”. The session was conducted in individual booths. Fermented milk beverages were evaluated on day 0 of storage, presented in three-digit blind codes, and 20 mL samples were served one at a time at 7 °C, simultaneously with a glass of water for mouth rinsing among samples. The sensory attributes were appearance, color, aroma, flavor, consistency, firmness, and overall impression [36].

The just-about-right (JAR) scale was used to identify the optimal intensity of each attribute (Li et al. [37]). The scale consists of 5 points from 1—much too little—to 5—much too much—with the central point of 3 being “Just About Right”. Aroma (acid, alcoholic, cupuassu, and milk), taste (sweet, acid, and bitter), color (white, brown, and beige), flavor (cupuassu and caramel), and texture (sandiness, consistency, firmness, viscosity, and mouthfeel) were evaluated [20,37].

Participants also performed CATA, using terms referenced in other studies [38,39,40] to identify product characteristics: flavor (whey, milk, sweetened, cocoa, and fermented), taste (acid and bitter), color (artificial, beige, white, dark brown, and light brown), aroma (milk, sweet, acid, buttermilk, cupuassu, cocoa, and alcoholic), and appearance (homogeneous, sandy, firm, consistent, viscous, liquid, and fibrous). Finally, purchase intention was evaluated using a 5-point structured scale (1—would certainly not buy; 5—would certainly buy).

### 2.8. Estimation of Harrell’s Optimism on Regression Models Using Bootstrap Method

The performance of a predictive model is overestimated (optimism) when only determined on the sample used to construct the model. Therefore, internal validation methods aim to accurately estimate model performance in new samples [41]. The estimation of optimism was calculated according to Equation (5) [42].
(6)$$o=\frac{\sum_{m=1}^{M}o^{\left(m\right)}}{M},$$
where for each bootstrap sample with replacement (*m* = 1, …, *M*), $R_{boot}^{2\left(m\right)}$ = bootstrap coefficient of determination obtained from the model fitted to the bootstrap dataset; $R_{orig}^{2\left(m\right)}$ = original coefficient of determination obtained by applying the fitted model from the bootstrap dataset to the original dataset; *o* = optimism of the original model; $o^{\left(m\right)}=R_{boot}^{2\left(m\right)}-R_{orig}^{2\left(m\right)}$; and *M* = number of bootstrap datasets.

Equation (6) was used to calculate the coefficient of determination of the original model after validation.
(7)$$R_{V}^{2}=R_{app}^{2}-o,$$
where $R_{V}^{2}$ = coefficient of determination of the original model, $R_{app}^{2}$ = apparent coefficient of determination obtained from model fitted to original data, and o = optimism of the original model.

### 2.9. Statistical Analysis

All analyses were performed in analytical and experimental triplicate, and the data obtained were presented as means ± standard deviations (SDs). The comparison of multiple samples was performed using one-way analysis of variance (ANOVA) for centesimal results and two-way ANOVA for physicochemical (pH, syneresis, water-holding capacity, instrumental color, and rheological characteristics) and microbial results, using GraphPad Prism 8 (San Diego, CA, USA). ANOVA was followed by Tukey’s multiple comparison tests (two-sided; *p <* 0.05). In addition, Pearson’s correlation test with a 0.05 significance level was performed to evaluate the correlation between variables. The internal validation of the regression models was performed with the bootstrap method (number of bootstrap samples = 200; number of simulations = 1000; bootstrap sample size = original sample size; 95% confidence interval) [42]. In CATA, Cochran’s Q-test was performed to identify significant differences in the frequency of terms used to describe the samples (*p <* 0.05). When significant values were found, Sheskin’s multiple pairwise comparison test was employed at a 0.05 significance level to identify significant differences in each sensory term between samples. Statistical analyses were performed with a commercially available statistical package, XLSTAT, version 2022.3.2.1353 (Addinsoft SARL, New York, NY, USA).

## 3. Results and Discussion

### 3.1. Proximate Composition

The moisture, ash, protein, fat, fiber, and carbohydrate contents in the artisanal fermented beverages are presented in Table 1. HPNF showed the highest moisture value, and LPHF, the lowest (*p <* 0.05), indicating that cupuassu pulp and flour addition directly interfered with this parameter. This fact can be attributed to the high moisture content in pulp [15] and the low moisture content in flour [19], which is attributed to the technological process difference between these two products.

The highest values (*p <* 0.05) of protein, fat, fiber, and carbohydrates were found in cupuassu flour samples. This pattern happened due to the concentration of nutrients resulting from the technological transformation to which the cupuassu seeds were subjected [18,19].

### 3.2. Physicochemical Behavior

#### 3.2.1. pH Behavior

The variations in pH values presented during the storage period of the artisanal fermented milk beverages are shown in Table 2. The addition of pulp significantly reduced the pH value (*p <* 0.05), which was evident when comparing HPNF (10% pulp and 0% flour) and NPNF (control sample). In this case, pulp addition was a predictor of pH values (R_v_ = 0.742; *p* < 0.0001; Appendix A; Figure 1), which could be linked to the acidic pH of the cupuassu pulp preparation (3.09 ± 0.01) used. Similar behaviors were presented by Costa et al. [11], who added 10% of pasteurized cupuassu pulp to goat milk yogurt, finding decreased pH after pulp addition. However, the pH values were higher (4.57–4.43) than those herein, which could be attributed to the matrices and pulp conditions.

In addition, it was observed that the pH of cupuassu flour (5.70 ± 0.02) presented a correlation (*p* ˂ 0.05) with the percentage of added flour, and the model had poor predictive power (R_v_ = 0.599), given that the higher the percentage of cupuassu flour was, the higher the pH value was, with NPHF (0% pulp and 3% flour) being the treatment with the highest value (*p <* 0.05). This behavior was also present in the samples with a combination of pulp and flour, whose pH values were intermediate, compared with the samples that contained only pulp or flour. According to Yadav et al. [43], the decrease or increase in pH values can vary due to the acidity of the flour used.

**Figure 1 foods-12-02217-f001:**
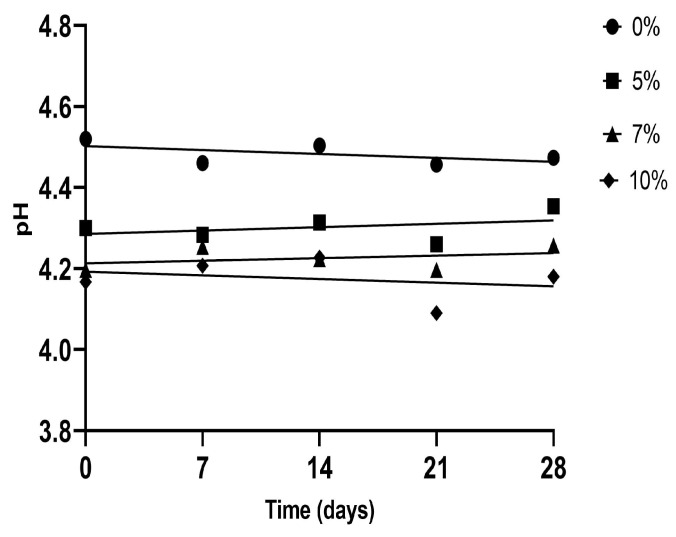
Significant correlations (*p <* 0.05) internally validated with the bootstrap method between pH and cupuassu pulp in artisanal fermented milk beverages stored at 4 °C.

During the storage period, NPNF and the samples with the addition of cupuassu flour showed a decrease in pH values, except for samples MPHF and LPHF combined with pulp and 3% cupuassu flour, which showed an increase in pH values between day 0 (4.25 and 4.44) and day 28 (4.42 and 4.52, respectively). This reduction could be attributed to the post-acidification generated in the dairy beverages by lactic acid bacteria (LAB) transforming lactose into lactic acid, allowing to maintain metabolic activity and acidification during storage under refrigeration [44]. The pH decrease of the sample with flour alone during storage could have been due to a possible synergistic effect towards starter cultures, attributed to fiber and protein contents in cupuassu flour, in addition to post-acidification, and concomitant protein/fiber catabolism, which caused the pH to slightly (although significantly) increase compared with the control. This behavior is similar to that found by Zare et al. [45] and Kaur et al. [46], who added lentil and soybean flour to fermented milk and chickpea flour in yogurts, respectively. It enhanced the acidification rate of probiotic cultures.

Furthermore, when comparing storage days 0 and 28, adding cupuassu pulp increased pH values (*p <* 0.05), except in the HPLF and MPLF treatments. On the other hand, it was observed that NPNF (control sample) had a decrease in pH values on days 0 and 21 of storage. While this is true, the treatments with a combination of cupuassu pulp and flour only showed this decrease on day 21, which indicates that the combination of pulp and flour in the beverage generated a slower post-acidification process, which would help to prolong the shelf life of the product [44] and which could have been due to the presence of the cupuassu yeasts (*Pichia* genus), which metabolize acid, causing the increase in pH [47]. Despite the observed alkalinization during storage, it was observed that the combination of cupuassu pulp and flour maintained pH values between 4.0 and 4.6, which is known to allow the development of LAB and prevent the growth of pathogenic microorganisms [19].

#### 3.2.2. Syneresis and Water-Holding Capacity (WHC)

The results of syneresis presented in Table 3 show that NPNF (control sample), in general, presented the highest water loss, regardless of the storage day (*p <* 0.05). The contraction of the casein network formed during fermentation justifies this behavior [48,49]. Cupuassu pulp and flour percentages were inversely correlated with syneresis values, that is, the greater the pulp or flour percentage was, the lower the syneresis values were, where HPHF (3% flour and 10% pulp) presented the lowest value (*p <* 0.05). Since syneresis is an indicator of quality, using these ingredients can mitigate this technological defect, which commonly affects the sensory parameters of this type of product. The reduced syneresis could be attributed to the pectin polysaccharides contained in cupuassu, which strengthen the casein network formed by interacting with the positive charges on the surface of micelles [14,50].

During storage, the control sample did not show any alterations (*p >* 0.05). However, the addition of HPNF pulp produced a slight increase, contrary to the addition of 3% flour alone, which decreased syneresis values (*p <* 0.05), presenting a predictor (R_v_ = 0.689; *p* < 0.0001; Appendix A) between flour percentage and syneresis (Figure 2). On the other hand, the values of HPLF increased during storage, while those of LPLF and LPHF decreased. In general, syneresis was not affected by storage time, and it was slight when there were significant differences. Thus, the addition of cupuassu did not appear to be a parameter relevant to syneresis with regard to storage days.

The results obtained for WHC confirmed the syneresis values and showed an inverse correlation between them, where syneresis was a strong predictor of WHC (R_v_ = 0.998; *p <* 0.0001; Appendix A; Figure 3a), since the low water-holding capacity of proteins increases whey separation in fermented milk beverages [51]. In addition, the percentages of pulp and flour used were found to be positively correlated with WHC (Figure 3b). In addition, similar to syneresis, WHC was slightly altered, if at all, by storage time. This behavior is related to the different pH values that occurred during the storage period, since low pH values are among the causes that generate the release of whey in a product [52].

#### 3.2.3. Instrumental Color

The instrumental color parameters of *L** (lightness), *a** (greenness–redness), *b** (blueness–yellowness), *C** (chroma), and *h°* (hue angle) of artisanal fermented milk drinks during the 28 days of storage are presented in Table 4. It is shown that adding pulp, in general, significantly reduced the *L** parameter (*p <* 0.05), except on day 0, where there was no significant difference from the control. This fact may be attributed to the pigments present in pulp, which make the beverage darker [13]. Likewise, the *L** values decreased more sharply in treatments with flour than in those with pulp. Thus, flour intensified the darkening when combined with pulp, as evidenced in LPHF (5% pulp and 3% flour), which exhibited the lowest value of this parameter (*p <* 0.05). Therefore, flour content was a strong predictor of *L** (R_v_ = 0.867; Figure 4), which resulted from the dark pigmentation of the dust due to the oxidation of phenolic compounds in cupuassu seeds during technological processing [18].

During storage, the *L** values progressively increased in all samples during the 28 days, except for NPNF, whose values only increased until day 7 (*p <* 0.05). The reduction in the size of the casein micelles due to proteolysis during storage can increase *L** values due to increased scattering light [53]. However, considering the treatments with added cupuassu pulp, the degradation of pigments during storage may have also contributed to this behavior [47].

**Table 4 foods-12-02217-t004:** Color parameters values of artisanal fermented milk beverages with cupuassu pulp and flour during refrigerated storage.

Parameter	Sample	Storage Time (Days)				
		0	7	14	21	28
** *L** **	NPNF	35.95 ± 0.88 ^abB^	37.41 ± 0.09 ^aA^	37.27 ± 0.01 ^aA^	37.74 ± 0.05 ^aA^	37.49 ± 0.10 ^bA^
	NPLF	14.41 ± 0.06 ^deD^	14.26 ± 0.01 ^hE^	14.74 ± 0.05 ^dC^	14.89 ± 0.03 ^hB^	15.34 ± 0.01 ^fA^
	NPHF	14.88 ± 0.50 ^cdeAB^	14.28 ± 0.03 ^hC^	14.68 ± 0.01 ^dBC^	14.89 ± 0.02 ^hAB^	15.31 ± 0.02 ^fA^
	LPNF	35.13 ± 0.06 ^bE^	35.29 ± 0.04 ^cD^	35.69 ± 0.09 ^bC^	35.94 ± 0.01 ^dB^	36.29 ± 0.02 ^cA^
	MPNF	36.80 ± 0.02 ^aAB^	35.58 ± 0.08 ^bB^	36.19 ± 1.04 ^bB^	37.53 ± 0.02 ^bA^	37.79 ± 0.05 ^aA^
	HPNF	35.56 ± 0.07 ^bC^	35.17 ± 0.03 ^dD^	35.64 ± 0.06 ^bC^	36.05 ± 0.04 ^cB^	36.23 ± 0.02 ^cA^
	HPLF	15.02 ± 0.01 ^cdeD^	15.87 ± 0.02 ^eC^	16.32 ± 0.02 ^cA^	16.20 ± 0.04 ^eB^	16.34 ± 0.01 ^eA^
	MPLF	15.32 ± 0.02 ^cD^	15.45 ± 0.01 ^fC^	14.94 ± 0.01 ^dE^	15.87 ± 0.01 ^fB^	16.72 ± 0.03 ^dA^
	LPLF	14.56 ± 0.25 ^cdeC^	14.77 ± 0.02 ^gBC^	15.02 ± 0.03 ^dAB^	15.10 ± 0.03 ^gA^	15.03 ± 0.01 ^gAB^
	HPHF	15.05 ± 0.01 ^cdeD^	15.86 ± 0.01 ^eC^	16.28 ± 0.01 ^cA^	16.21 ± 0.04 ^eB^	16.31 ± 0.01 ^eA^
	MPHF	15.28 ± 0.01 ^cdD^	15.44 ± 0.01 ^fC^	14.94 ± 0.01 ^dE^	15.86 ± 0.01 ^fB^	16.67 ± 0.01 ^dA^
	LPHF	14.26 ± 0.02 ^eE^	14.79 ± 0.02 ^gD^	14.98 ± 0.01 ^dC^	15.10 ± 0.02 ^gA^	15.02 ± 0.01 ^gB^
** *a** **	NPNF	1.140 ± 0.12 ^fA^	0.530 ± 0.01 ^iC^	0.6567 ± 0.08 ^gC^	0.570 ± 0.02 ^iC^	0.880 ± 0.05 ^iB^
	NPLF	1.897 ± 0.01 ^dC^	1.970 ± 0.02 ^dB^	2.130 ± 0.02 ^dA^	1.817 ± 0.01 ^dD^	1.915 ± 0.60 ^dC^
	NPHF	1.857 ± 0.01 ^dA^	1.877 ± 0.02 ^eA^	1.757 ± 0.03 ^eB^	1.777 ± 0.05 ^deB^	1.915 ± 0.01 ^dA^
	LPNF	2.177 ± 0.03 ^cE^	2.447 ± 0.01 ^cA^	2.407 ± 0.02 ^cB^	2.320 ± 0.02 ^cC^	2.220 ± 0.01 ^cD^
	MPNF	2.590 ± 0.03 ^bC^	2.757 ± 0.06 ^bB^	2.867 ± 0.01 ^bA^	2.600 ± 0.01 ^bC^	2.595 ± 0.04 ^bC^
	HPNF	3.347 ± 0.08 ^aC^	3.590 ± 0.03 ^aB^	3.647 ± 0.02 ^aAB^	3.710 ± 0.01 ^aA^	3.360 ± 0.01 ^aC^
	HPLF	1.560 ± 0.03 ^eC^	1.730 ± 0.04 ^ghB^	1.620 ± 0.01 ^fC^	1.717 ± 0.01 ^efB^	1.815 ± 0.01 ^eA^
	MPLF	1.847 ± 0.03 ^dB^	1.977 ± 0.02 ^dA^	1.630 ± 0.01 ^fD^	1.697 ± 0.04 ^fgC^	1.645 ± 0.02 ^fCD^
	LPLF	1.817 ± 0.01 ^dA^	1.867 ± 0.02 ^efA^	1.830 ± 0.04 ^eA^	1.590 ± 0.01 ^hB^	1.560 ± 0.01 ^ghB^
	HPHF	1.577 ± 0.02 ^eD^	1.690 ± 0.02 ^gB^	1.630 ± 0.03 ^fC^	1.670 ± 0.01 ^fgBC^	1.760 ± 0.01 ^eA^
	MPHF	1.877 ± 0.01 ^dA^	1.917 ± 0.03 ^deA^	1.580 ± 0.01 ^fC^	1.730 ± 0.03 ^efB^	1.620 ± 0.02 ^fgC^
	LPHF	1.770 ± 0.02 ^dA^	1.800 ± 0.01 ^fgA^	1.787 ± 0.01 ^eA^	1.637 ± 0.04 ^ghB^	1.500 ± 0.05 ^hC^
** *b** **	NPNF	26.290 ± 1.26 ^aA^	26.997 ± 0.14 ^aA^	26.677 ± 0.19 ^bA^	26.810 ± 0.18 ^aA^	26.455 ± 0.03 ^aA^
	NPLF	−2.340 ± 0.02 ^bC^	−2.020 ± 0.01 ^eA^	−2.090 ± 0.02 ^eA^	−2.290 ± 0.01 ^defgC^	−2.215 ± 0.06 ^fB^
	NPHF	−2.210 ± 0.08 ^bBC^	−2.000 ± 0.02 ^eA^	−2.310 ± 0.04 ^fgC^	−2.270 ± 0.04 ^defC^	−2.145 ± 0.01 ^eB^
	LPNF	26.300 ± 0.15 ^aA^	26.327 ± 0.13 ^bA^	26.180 ± 0.07 ^cAB^	25.970 ± 0.03 ^cB^	26.175 ± 0.02 ^bAB^
	MPNF	26.820 ± 0.01 ^aB^	25.950 ± 0.04 ^cD^	27.650 ± 0.02 ^aA^	26.397 ± 0.01 ^bC^	25.495 ± 0.02 ^cE^
	HPNF	26.557 ± 0.12 ^aA^	25.270 ± 0.08 ^aC^	25.267 ± 0.06 ^dC^	25.897 ± 0.01 ^cB^	25.195 ± 0.03 ^dC^
	HPLF	−2.537 ± 0.04 ^bC^	−2.380 ± 0.05 ^fgB^	−2.447 ± 0.03 ^ghB^	−2.247 ± 0.01 ^deA^	−2.260 ± 0.01 ^fA^
	MPLF	−2.327 ± 0.02 ^bB^	−2.167 ± 0.02 ^efA^	−2.590 ± 0.01 ^hD^	−2.300 ± 0.01 ^defgB^	−2.440 ± 0.01 ^gC^
	LPLF	−2.177 ± 0.02 ^bA^	−2.190 ± 0.03 ^efA^	−2.227 ± 0.08 ^efA^	−2.450 ± 0.02 ^gB^	−2.725 ± 0.01 ^hC^
	HPHF	−2.600 ± 0.01 ^bC^	−2.410 ± 0.02 ^fgB^	−2.400 ± 0.02 ^fghB^	−2.237 ± 0.015 ^dA^	−2.250 ± 0.01 ^fA^
	MPHF	−2.410 ± 0.19 ^bB^	−2.070 ± 0.01 ^eA^	−2.537 ± 0.01 ^hB^	−2.417 ± 0.01 ^fgB^	−2.420 ± 0.02 ^gB^
	LPHF	−2.167 ± 0.02 ^bA^	−2.507 ± 0.23 ^gBC^	−2.230 ± 0.01 ^efAB^	−2.410 ± 0.04 ^efgABC^	−2.685 ± 0.03 ^hC^
** *C** **	NPNF	26.317 ± 1.27 ^aA^	27.007 ± 0.14 ^aA^	26.687 ± 0.19 ^bA^	26.817 ± 0.19 ^aA^	26.470 ± 0.02 ^aA^
	NPLF	3.010 ± 0.010 ^bA^	2.850 ± 0.04 ^eB^	2.980 ± 0.01 ^efA^	2.930 ± 0.01 ^deAB^	2.930 ± 0.07 ^fgAB^
	NPHF	2.887 ± 0.07 ^bA^	2.757 ± 0.03 ^eB^	2.900 ± 0.02 ^efA^	2.887 ± 0.06 ^deA^	2.875 ± 0.01 ^fgA^
	LPNF	26.387 ± 0.15 ^aA^	26.440 ± 0.12 ^bA^	26.297 ± 0.07 ^cAB^	26.077 ± 0.04 ^cB^	26.275 ± 0.02 ^bAB^
	MPNF	26.947 ± 0.01 ^aB^	26.097 ± 0.04 ^cD^	27.800 ± 0.02 ^aA^	26.520 ± 0.01 ^bC^	25.625 ± 0.02 ^cE^
	HPNF	26.760 ± 0.12 ^aA^	25.527 ± 0.09 ^dC^	25.527 ± 0.06 ^dC^	26.167 ± 0.01 ^cB^	25.415 ± 0.03 ^dC^
	HPLF	2.977 ± 0.05 ^bA^	2.947 ± 0.07 ^eA^	2.930 ± 0.02 ^efA^	2.827 ± 0.01 ^deB^	2.900 ± 0.01 ^fgAB^
	MPLF	2.967 ± 0.01 ^bB^	2.930 ± 0.02 ^eC^	3.060 ± 0.01 ^eA^	2.857 ± 0.02 ^deD^	2.945 ± 0.01 ^fBC^
	LPLF	2.830 ± 0.01 ^bC^	2.877 ± 0.04 ^eBC^	2.880 ± 0.03 ^efBC^	2.920 ± 0.02 ^deB^	3.135 ± 0.01 ^eA^
	HPHF	3.047 ± 0.01 ^bA^	2.947 ± 0.01 ^eB^	2.907 ± 0.04 ^efB^	2.790 ± 0.02 ^eD^	2.855 ± 0.01 ^gC^
	MPHF	2.920 ± 0.01 ^bB^	2.820 ± 0.01 ^eC^	2.987 ± 0.01 ^efA^	2.970 ± 0.01 ^dA^	2.910 ± 0.03 ^fgB^
	LPHF	2.807 ± 0.03 ^bC^	2.827 ± 0.05 ^eBC^	2.857 ± 0.02 ^fBC^	2.917 ± 0.06 ^deB^	3.075 ± 0.05 ^eA^
** *h°* **	NPNF	87.530 ± 0.15 ^dC^	88.870 ± 0.01 ^fA^	88.610 ± 0.14 ^fA^	88.777 ± 0.04 ^fA^	88.090 ± 0.11 ^gB^
	NPLF	308.987 ± 0.34 ^abD^	313.780 ± 0.24 ^aB^	315.610 ± 0.53 ^aA^	308.417 ± 0.08 ^aD^	310.865 ± 0.19 ^bC^
	NPHF	309.980 ± 0.95 ^aAB^	311.520 ± 1.44 ^bcA^	307.080 ± 0.94 ^cC^	308.027 ± 0.21 ^aBC^	311.760 ± 0.01 ^aA^
	LPNF	85.270 ± 0.03 ^eA^	84.690 ± 0.01 ^gD^	84.750 ± 0.04 ^gD^	84.887 ± 0.04 ^gC^	85.155 ± 0.01 ^hB^
	MPNF	84.477 ± 0.07 ^eA^	83.947 ± 0.13 ^gD^	84.090 ± 0.01 ^gCD^	84.370 ± 0.02 ^gAB^	84.190 ± 0.07 ^iBC^
	HPNF	82.820 ± 0.12 ^fA^	81.917 ± 0.05 ^hC^	81.790 ± 0.02 ^hC^	81.850 ± 0.01 ^hC^	82.410 ± 0.01 ^jB^
	HPLF	301.647 ± 0.17 ^cE^	306.017 ± 0.07 ^eC^	303.520 ± 0.14 ^deD^	307.310 ± 0.05 ^bB^	308.775 ± 0.05 ^cA^
	MPLF	308.407 ± 0.57 ^bB^	312.410 ± 0.02 ^abA^	302.137 ± 0.05 ^eE^	306.420 ± 0.72 ^cC^	304.020 ± 0.38 ^eD^
	LPLF	309.817 ± 0.21 ^aA^	310.450 ± 0.08 ^cdA^	309.457 ± 1.64 ^bA^	302.970 ± 0.05 ^eB^	299.800 ± 0.02 ^fC^
	HPHF	301.257 ± 0.40 ^cE^	305.097 ± 0.52 ^eC^	304.150 ± 0.23 ^dD^	306.817 ± 0.02 ^bcB^	307.985 ± 0.24 ^dA^
	MPHF	309.907 ± 0.35 ^aB^	312.787 ± 0.49 ^abA^	301.950 ± 0.10 ^eE^	306.207 ± 0.05 ^cC^	303.860 ± 0.12 ^eD^
	LPHF	309.267 ± 0.11 ^abA^	309.540 ± 0.54 ^dA^	308.657 ± 0.02 ^bcA^	304.080 ± 0.11 ^dB^	299.235 ± 0.53 ^fB^
**Δ*E***	NPNF	NA	2.03 ± 1.10 ^aAB^	1.89 ± 0.86 ^aAB^	2.26 ± 1.00 ^aA^	1.89 ± 0.72 ^aAB^
	NPLF	NA	0.37 ± 0.01 ^cdB^	0.47 ± 0.01 ^bcB^	0.48 ± 0.09 ^bB^	0.93 ± 0.06 ^bcdA^
	NPHF	NA	0.68 ± 0.38 ^bcdA^	0.42 ± 0.27 ^cA^	0.36 ± 0.23 ^bA^	0.50 ± 0.43 ^dA^
	LPNF	NA	0.32 ± 0.01 ^dD^	0.62 ± 0.04 ^bcC^	0.90 ± 0.02 ^bB^	1.18 ± 0.06 ^abcdA^
	MPNF	NA	1.51 ± 0.08 ^abA^	1.29 ± 0.50 ^abB^	0.84 ± 0.03 ^bB^	1.66 ± 0.01 ^abA^
	HPNF	NA	1.37 ± 0.17 ^abcA^	1.33 ± 0.08 ^abA^	0.91 ± 0.02 ^bB^	1.52 ± 0.08 ^abA^
	HPLF	NA	0.89 ± 0.01 ^bcdD^	1.30 ± 0.02 ^abB^	1.22 ± 0.03 ^bC^	1.37 ± 0.01 ^abcA^
	MPLF	NA	0.24 ± 0.02 ^dAD^	0.51 ± 0.01 ^bcC^	0.57 ± 0.01 ^bB^	1.42 ± 0.01 ^abcA^
	LPLF	NA	0.23 ± 0.21 ^dBC^	0.46 ± 0.24 ^bcABC^	0.66 ± 0.22 ^bAB^	0.79 ± 0.14 ^cdA^
	HPHF	NA	0.84 ± 0.01 ^bcdC^	1.25 ± 0.01 ^abB^	1.22 ± 0.04 ^bB^	1.33 ± 0.01 ^abcA^
	MPHF	NA	0.39 ± 0.18 ^cdC^	0.49 ± 0.06 ^bcBC^	0.62 ± 0.02 ^bB^	1.42 ± 0.02 ^abcA^
	LPHF	NA	0.66 ± 0,18 ^bcdC^	0.72 ± 0.02 ^bcBC^	0.89 ± 0.01 ^bAB^	0.96 ± 0.01 ^bcdA^

^a–j^ Different lowercase superscripts in the same column indicate significant differences among treatments of artisanal fermented milk beverage (*p <* 0.05). ^A–E^ Different uppercase superscripts in the same row indicate significant differences among storage times (*p <* 0.05). NPNF, control; NPLF, beverage (1.5% flour); NPHF, beverage (3% flour); LPNF, beverage (5% pulp); MPNF, beverage (7.5% pulp); HPNF, beverage (10% pulp); HPLF, beverage (10% pulp and 1.5% flour); MPLF, beverage (7.5% pulp and 1.5% flour); LPLF, beverage (5% pulp and 1.5% flour); HPHF, beverage (10% pulp and 3% flour); MPHF, beverage (7.5% pulp and 3% flour); LPHF, beverage (5% pulp and 3% flour); *L**, lightness; *a**, redness; *b**, yellowness; *C**, chroma; and *h°*, hue angle; Δ*E**, total color difference.

The addition of cupuassu increased the values of *a** compared with the control. However, the values obtained with pulp were higher than those obtained with flour due to the reddish color developed by cupuassu catechins due to oxidation, a product of fermentation [22,24]. Coherently, there was a positive correlation between *a** and pulp concentration, while a negative correlation between *a** and flour was observed in the beverages (Appendix A). Thus, treatments combining pulp and flour presented higher *a** values than the control but lower values than those with pulp alone. Regarding storage time, NPNF showed a noticeable decrease during this period (*p <* 0.05), which could be associated with the pH behavior of the samples during storage being a low predictor of the *a** parameter (*p* < 0.0001). On the other hand, in samples containing cupuassu, the values of *a** remained unchanged, increased, or decreased. This could be attributed to the influence of pH, as well as the stability of catechin, on each treatment.

The *b** values were significantly higher in the control (*p <* 0.05) and in the treatments with pulp alone, regardless of storage day. It was attributed to the yellow–green color of whey due to the presence of riboflavin and the yellowish white of cupuassu pulp [6,18]. The control had a stable behavior during the 28 days of storage. In contrast, treatments with flour or pulp alone had reduced or unchanged *b** values. An increase in *b** values is typical of non-enzymatic browning (Maillard) reactions [54]. Thus, flour and pulp apparently mitigated the Maillard reaction. Phenolic compounds, which are present in cupuassu, suppress the development of Maillard browning compounds in foods [25]. On the other hand, flour and pulp had the opposite effect when combined, raising *b** values. It is possible that the carbohydrate and protein contents increased with the two ingredients (Table 1) superimposed the suppressive effect of phenolic compounds.

The chroma (*C**) results indicated that treatments with pulp alone presented *c** values similar to or slightly lower than those in the control. In contrast, samples with flour alone sharply decreased chroma, including when combined with pulp (Figure 5a). Therefore, the lowest color saturation in samples containing flour can be attributed to its effect of decreasing *L** and *b** compared with the control, which means that flour content was a strong predictor of *C** values (R_v_ = 0.869), as reported in Appendix A and Figure 5b. Comparing the samples at the end of storage and those freshly prepared, control and flour treatments remained stable regarding chroma. In general, samples with pulp alone had reduced color saturation, while the values of combined treatments were similar to or higher than those of the control, such as LPLF and LPHF. Chroma behavior during storage was similar to that of *b**, which can be explained by the significant correlation between them (*p <* 0.001), as shown in Appendix A.

Hue angle (*h°*) values identify the relative orientation of the color with respect to the origin as 0° (red-purple), 90° (yellow), 180° (blue-green), and 270° (blue) to specify the color. The *h°* values obtained indicated that the values of pulp-only samples were significantly lower than those of the control, presenting a yellow color perception (*p <* 0.05). On the other hand, the samples containing flour showed an increase in the value of *h°* (Figure 6a) associated with the magenta color, which could be attributed to the pigmentation of the flour and the reduction in syneresis, allowing a better perception of the hue, the latter generated by the increase in pH and WHC, positively influencing the decrease in *L** (Figure 6b), *b**, and *C** compared to the control, which explains the significant correlation with the values of *h°* (R_v_ = 868; *p* < 0.0001; Appendix A).

On day 28 of storage, the samples only containing flour and the control showed an increase in *h°* values compared with day 0. On the contrary, the addition of pulp caused a decrease in *h°*. The same behavior was observed in the combined treatments with pulp and flour, with which the values decreased, except for HPLF and HPHF, with which the values increased at the end of storage. The behavior of *h*° during storage was similar to that of *a**, indicating a significant correlation between them (*p =* 0.001; Appendix A).

Δ*E* indicates the color stability present in samples. Samples LPNF, NPLF, and NPHF showed significantly lower values (*p <* 0.05) than the control. The same behavior was found in the combined samples of pulp and flour, where MPLF and LPLF showed significantly lower values (*p <* 0.05), indicating that higher percentages of pulp and flour resulted in a more stable color during storage, which is excellent from the consumer’s point of view. During the remaining days of storage (14, 21, and 28), Δ*E* values increased in all samples except NPHF and NPNF, which showed no significant difference during storage (*p >* 0.05).

### 3.3. Rheological Behavior

The behavior of the fermented dairy beverages with added cupuassu pulp and flour was described using the Ostwald–de Waele model, as reported in Table 5. This model relates the shear stress to the shear rate of a fluid [33]. According to Moreira et al. [55], it provides better adjustment to the experimental data for whey-based products, similar to that reported herein. The values obtained for the correlation coefficient in the different samples were greater than 0.910 (R^2^ ≥ 0.910), which indicates that the power law model provided suitable adjustment parameters. The consistency index (K) gives an idea of the fluid viscosity; the K values were generally unaffected by flour alone compared with the control, regardless of the added concentration (*p >* 0.05). On the other hand, the medium and high pulp concentrations elevated the K values, while the low concentration was insignificant. Similarly, 10% and 7.5% pulp in combined treatments provided higher consistency indexes than the control, regardless of flour concentration. In contrast, 5% pulp generally did not affect K, regardless of the combined flour concentration. The addition of pulp presented a positive correlation with K. Pulp acidified the pH and reduced syneresis, mainly when at a higher concentration. Accordingly, these factors contributed to increased K, as reported by the inverse correlation between them (Appendix A). However, increased WHC obtained by adding pulp also favored higher consistency (*p =* 0.03; Appendix A).

At the end of storage, despite fluctuations, the control and the treatments with flour alone and with pulp alone at 10% had K values similar to those of the freshly prepared product. In contrast, pulp alone at the medium and low concentrations progressively increased the consistency index. The behavior was varied for combined treatments with 3% flour; HPLF and MPLF had lower and higher final values than the control, respectively, while LPLF was similar to the control after 28 days in terms of values. In combined treatments with 1.5% flour, the medium and low pulp concentrations elevated K during storage, while the high pulp concentration resulted in K values similar to those of the control. These differences in behavior can be attributed to the different effects of treatments on pH, syneresis, and WHC during storage (Appendix A).

**Table 5 foods-12-02217-t005:** Rheological parameters obtained with Ostwald–de Waele model of artisanal fermented milk beverages.

Parameter	Sample	Storage Time (Days)				
		0	7	14	21	28
**K (mPa·s^n^)**	NPNF	230.59 ± 41.49 ^efC^	208.88 ± 35.18 ^gC^	351.11 ± 17.01 ^eAB^	402.64 ± 14.38 ^dA^	283.49 ± 65.70 ^efBC^
	NPLF	161.89 ± 1.94 ^fC^	198.50 ± 47.44 ^gBC^	385.46 ± 39.57 ^eA^	290.75 ± 49.20 ^eAB^	235.03 ± 2.51 ^fBC^
	NPHF	413.84 ± 47.67 ^cdA^	166.83 ± 15.96 ^gC^	306.27 ± 65.92 ^eB^	388.41 ± 20.70 ^dAB^	357.08 ± 24.35 ^eAB^
	LPNF	174.29 ± 44.94 ^efD^	252.67 ± 46.45 ^fgCD^	365.38 ± 5.42 ^eBC^	449.35 ± 39.63 ^dAB^	530.60 ± 61.26 ^dA^
	MPNF	406.37 ± 19.27 ^cdD^	368.59 ± 30.99 ^deD^	543.88 ± 11.13 ^cdC^	774.28 ± 1.07 ^bB^	1284.53 ± 57.90 ^aA^
	HPNF	556.90 ± 1.95 ^cA^	574.04 ± 54.25 ^abA^	650.88 ± 38.77 ^bA^	640.47 ± 22.88 ^cA^	563.23 ± 44.25 ^dA^
	HPLF	1699.10 ± 146.40 ^aA^	502.12 ± 23.98 ^bcC^	663.73 ± 13.83 ^bC^	944.51 ± 50.64 ^aB^	530.26 ± 23.16 ^dC^
	MPLF	390.43 ± 63.37 ^dC^	625.36 ± 43.18 ^aB^	581.54 ± 15.67 ^bcB^	625.30 ± 52.86 ^cB^	949.46 ± 7.44 ^bA^
	LPLF	176.59 ± 2.52 ^efC^	213.04 ± 50.82 ^gC^	331.69 ± 27.51 ^eAB^	407.97 ± 14.13 ^dA^	253.21 ± 32.37 ^efBC^
	HPHF	843.50 ± 5.47 ^bA^	473.90 ± 38.64 ^bcdC^	769.92 ± 26.69 ^aB^	849.56 ± 16.26 ^abA^	797.68 ± 27.82 ^cAB^
	MPHF	477.13 ± 34.40 ^cdC^	453.05 ± 35.28 ^cdeC^	490.73 ± 3.52 ^dC^	664.53 ± 16.36 ^cB^	926.42 ± 19.85 ^bA^
	LPHF	331.82 ± 36.22 ^deB^	356.20 ± 16.87 ^efB^	313.91 ± 13.26 ^eB^	449.45 ± 37.20 ^dA^	496.54 ± 4.63 ^dA^
**n**	NPNF	0.438 ± 0.03 ^bcdA^	0.461 ± 0.07 ^abA^	0.398 ± 0.03 ^bcA^	0.382 ± 0.02 ^abA^	0.426 ± 0.05 ^bcA^
	NPLF	0.458 ± 0.02 ^bcA^	0.454 ± 0.04 ^abA^	0.333 ± 0.01 ^deB^	0.405 ± 0.03 ^aA^	0.444 ± 0.01 ^abA^
	NPHF	0.385 ± 0.02 ^deB^	0.450 ± 0.02 ^abA^	0.398 ± 0.04 ^bcAB^	0.349 ± 0.02 ^bcdB^	0.379 ± 0.05 ^cdB^
	LPNF	0.488 ± 0.05 ^abA^	0.479 ± 0.03 ^aA^	0.417 ± 0.01 ^abAB^	0.351 ± 0.02 ^bcdB^	0.363 ± 0.02 ^deB^
	MPNF	0.350 ± 0.01 ^efB^	0.416 ± 0.02 ^abA^	0.359 ± 0.01 ^cdB^	0.309 ± 1 × 10^−3 dC^	0.229 ± 2 × 10^−3 gD^
	HPNF	0.348 ± 0.01 ^efB^	0.383 ± 0.01 ^bcA^	0.378 ± 2 × 10^−3 bcdA^	0.375 ± 3 × 10^−3 abA^	0.383 ± 0.01 ^cdA^
	HPLF	0.297 ± 0.01 ^fC^	0.401 ± 0.01 ^abA^	0.349 ± 2 × 10^−3 deB^	0.312 ± 0.01 ^cdC^	0.401 ± 0.02 ^bcdA^
	MPLF	0.397 ± 0.30 ^cdeA^	0.310 ± 0.01 ^cB^	0.307 ± 0.01 ^eB^	0.342 ± 0.01 ^bcdB^	0.308 ± 0.01 ^fB^
	LPLF	0.534 ± 0.01 ^aA^	0.458 ± 0.04 ^bcA^	0.413 ± 0.02 ^abCD^	0.384 ± 0.01 ^abD^	0.482 ± 0.02 ^aAB^
	HPHF	0.359 ± 0.03 ^efB^	0.434 ± 0.02 ^abA^	0.343 ± 3 × 10^−3 deB^	0.348 ± 2 × 10^−3 bcdB^	0.360 ± 0.01 ^defB^
	MPHF	0.375 ± 0.01 ^deB^	0.427 ± 1 × 10^−3 abA^	0.371 ± 3 × 10^−3 bcdB^	0.358 ± 0.01 ^abcC^	0.322 ± 1 × 10^−3 efD^
	LPHF	0.402 ± 0.01 ^cdeBC^	0.412 ± 0.02 ^abAB^	0.457 ± 0.01 ^aA^	0.360 ± 0.03 ^abcC^	0.382 ± 0.01 ^cdBC^
**Apparent viscosity (mPa∙s)**	NPNF	57.12 ± 15.78 ^efB^	49.50 ± 19.00 ^eB^	98.06 ± 12.77 ^fgAB^	118.04 ± 12.10 ^efA^	74.05 ± 27.57 ^fAB^
	NPLF	37.10 ± 3.04 ^deC^	47.34 ± 17.20 ^eBC^	132.32 ± 19.19 ^efA^	80.17 ± 20.10 ^fB^	56.35 ± 1.49 ^fBC^
	NPHF	119.56 ± 6.17 ^efA^	39.39 ± 5.72 ^eB^	87.07 ± 29.14 ^gA^	126.60 ± 14.28 ^efA^	105.60 ± 12.57 ^efA^
	LPNF	37.45 ± 15.04 ^eD^	54.82 ± 14.61 ^deCD^	95.50 ± 2.54 ^fgBC^	146.00 ± 22.78 ^deAB^	165.7 ± 31.10 ^dA^
	MPNF	131.90 ± 8.56 ^deCD^	97.25 ± 15.15 ^cdD^	171.07 ± 5.73 ^cdC^	286.56 ± 2.23 ^bB^	614.21 ± 32.71 ^aA^
	HPNF	181.54 ± 3.04 ^cdA^	167.47 ± 18.40 ^bA^	191.70 ± 13.36 ^bcA^	191.42 ± 7.67 ^cdA^	164.28 ± 17.93 ^dA^
	HPLF	655.25 ± 77.48 ^aA^	138.27 ± 11.04 ^bcC^	216.09 ± 7.25 ^bC^	345.76 ± 24.14 ^aB^	146.11 ± 12.68 ^deC^
	MPLF	110.35 ± 28.32 ^bcC^	230.93 ± 21.90 ^aB^	216.52 ± 9.52 ^bB^	208.23 ± 24.42 ^cB^	352.67 ± 5.87 ^bA^
	LPLF	31.63 ± 1.27 ^fC^	49.95 ± 17.31 ^eC^	88.01 ± 12.18 ^gB^	118.46 ± 3.33 ^efA^	54.01 ± 10.27 ^fC^
	HPHF	266.20 ± 19.98 ^bA^	117.42 ± 15.169 ^cB^	255.02 ± 5.60 ^aA^	277.42 ± 7.27 ^bA^	250.37 ± 10.14 ^cA^
	MPHF	142.52 ± 8.35 ^deC^	114.67 ± 8.48 ^cD^	148.76 ± 3.13 ^deC^	209.97 ± 5.37 ^cB^	328.24 ± 8.78 ^bA^
	LPHF	91.18 ± 12.57 ^efB^	94.87 ± 10.05 ^cdB^	72.34 ± 6.04 ^gB^	142.43 ± 25.40 ^eA^	145.34 ± 3.42 ^deA^

^a–g^ Different lowercase superscripts in the same column indicate significant differences among treatments of artisanal fermented milk beverage (*p <* 0.05). ^A–D^ Different uppercase superscripts in the same row indicate significant differences among storage times (*p <* 0.05). NPNF, control; NPLF, beverage (1.5% flour); NPHF, beverage (3% flour); LPNF, beverage (5% pulp); MPNF, beverage (7.5% pulp); HPNF, beverage (10% pulp); HPLF, beverage (10% pulp and 1.5% flour); MPLF, beverage (7.5% pulp and 1.5% flour); LPLF, beverage (5% pulp and 1.5% flour); HPHF, beverage (10% pulp and 3% flour); MPHF, beverage (7.5% pulp and 3% flour); LPHF, beverage (5% pulp and 3% flour); K (mPa s^n^), consistency index; n, flow behavior index; R^2^, linear correlation coefficient; apparent viscosity (measured at 25 °C at a 25 s^−1^ shear rate).

The n index indicates the degree of deviation from the Newtonian flow (n = 1). As n values were ≤ 0.534 (Table 5), all samples presented a non-Newtonian behavior. It indicates that the beverages had the typical performance of pseudoplastic fluids, which is characteristic of this type of dairy product [56,57]. This behavior is corroborated by Appendix A, where the shear stress increased as a function of the shear rate in all samples, regardless of storage time. The values of n were not affected by the addition of pulp alone at the low concentration compared with the control (*p >* 0.05), while the medium and high concentrations of pulp decreased the values of n (*p <* 0.05). On the other hand, the addition of flour did not generally affect the values of n, regardless of the added concentration (*p >* 0.05). Differently combined samples with the high pulp concentration provided lower consistency indices than the control, regardless of flour concentration, while samples with the medium pulp concentration showed no significant difference, regardless of flour concentration, as did LPHF (containing 5% pulp and 3% flour) in contrast to LPLF (containing the same concentration of pulp but less flour), with which the index was significantly higher (*p <* 0.05).

During storage, the control showed no significant difference (*p >* 0.05); in contrast to the samples with only flour and only pulp, LPNF and MPNF decreased n values, while HPNF increased with time, indicating that storage time, pulp addition, and pH favored n (Appendix A). The HPLF and MPHF combined samples showed an increase in n on day 7 of storage and a subsequent decrease until days 21 and 28, respectively. The value of n in MPLF decreased during storage, while that in LPLF only decreased after day 7. HPHF and LPHF presented the highest values on days 7 and 14, respectively. Consistently, K was a strong predictor of n (R_v_ = 0.786; *p* < 0.001; Appendix A); as the K values increased, the n values decreased (Figure 7a).

Apparent viscosity describes the flow behavior, an important parameter that influences sensory attributes and quality in dairy beverages, since the addition of whey generates low viscosity in this type of product [57]. Table 5 presents the values obtained for viscosity; it was observed that the addition of pulp at the highest concentration increased the apparent viscosity compared with the control (*p <* 0.05), while flour did not generate a significant effect, regardless of concentration (*p >* 0.05). On the other hand, the combination of flour and pulp ingredients in the LPLF, MPHF, and LPHF samples did not show significant changes compared with the control (*p >* 0.05), in contrast to HPLF, MPLF, and HPHF, which showed a significant increase in apparent viscosity values (*p >* 0.05).

During storage, the control sample showed an increase in viscosity on days 21 and 28; this behavior was also present in the samples with 1.5% flour, while the sample with 3% flour only showed a significant variation on day 7, obtaining the lowest value (*p <* 0.05). Samples with only-pulp addition increased viscosity over time, which could be attributed to the reduction in pH due to the addition of pulp favoring higher apparent viscosity (*p =* 0.04; Appendix A). Samples combined with the lowest pulp concentration (LPLF and LPHF) increased apparent viscosity values on days 21 and 28, respectively (*p <* 0.05). However, LPLF showed no significant difference from the control on day 28 (*p >* 0.05). Concerning the samples combined with the medium pulp concentration (MPHF and MPLF), they showed a decrease on day 7, where MPHF subsequently increased until day 28 (*p <* 0.05), showing a higher apparent viscosity value than the control. In general, HPLF and HPHF samples had the highest apparent viscosity at the beginning of storage (*p <* 0.05). However, these had the lowest values on day 7, which can be attributed to the decrease in n during storage and the increase in K caused by the addition of pulp, which positively affected the viscosity (Figure 7b; *p <* 0.001; Appendix A).

**Figure 7 foods-12-02217-f007:**
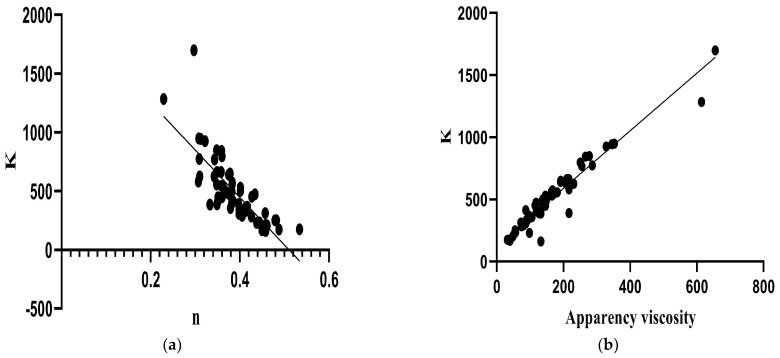
Significant correlations (*p <* 0.05) internally validated with the bootstrap method (**a**) between K and n and (**b**) between K and apparent viscosity in artisanal fermented milk beverages stored at 4 °C.

### 3.4. Microbiological Analysis

In the count of molds and yeasts on day 0 of storage, the lowest values were found in the LPLF and NPNF samples, with counts ≤ 4 log CFU/mL, while HPLF had the highest value, with 5.83 ± 0.08 log CFU/mL (*p <* 0.05). On day 28, samples MPNF, LPHF, HPLF, NPHF, and NPLF obtained the highest values (between 6.11 ± 0.12 and 6.20 ± 0.32), while NPNF had the lowest value, with 5.57 ± 0.13 (*p >* 0.05). Therefore, treatments without or with lower cupuassu content had less fungal growth than treatments with higher content. This could be attributed to the yeast (*Pichia* and *Hanseniaspora*) content in the cupuassu fruit microbiota and organic compounds in cupuassu pulp and seeds [47]. Regarding time, all counts significantly increased until the end of storage (*p <* 0.05), except those in HPHF and LPLF, which remained stable. The relatively high pH and water activity of artisanal fermented milk beverages favor the growth of this type of microorganism.

Total and thermotolerant coliforms, *Escherichia coli*, coagulase-positive *Staphylococcus*, and *Salmonella* spp. were absent in the artisanal fermented milk beverages during storage, indicating good hygienic and sanitary practices during the production and storage of the beverages.

### 3.5. Sensory Analysis

The results of the acceptability test and purchase intention of the artisanal fermented milk beverages with cupuassu flour and pulp are shown in Table 6 and Appendix A. Formulations with 1.5% flour tended to present lower sensory scores than formulations with 3% flour regarding appearance, color, consistency, and firmness. Thus, MPLF and LPLF scored lower than HPHF, while only HPLF scored lower than MPHF and LPHF in appearance. In terms of color, HPHF was superior to all treatments with 1.5% flour, while MPHF and LPHF scored higher than MPLF and LPLF. MPLF scored lower than all treatments with 3% flour in consistency, and it scored lower than HPHF and LPHF in firmness. The high flour concentration positively affected the appearance (R_v_ = 0.825; *p =* 0.032; Appendix A), color (R_v_ = 0.936; *p =* 0.005; Appendix A), and flavor (R_v_ = 0.803; *p =* 0.039) of the artisanal fermented milk beverages, which was attributed to the decrease in syneresis caused by the increase in WHC, which increased the scores in appearance and color attributes (Appendix A–D), as well as color and carbohydrate content typical of cupuassu flour [19]. On the other hand, flour concentration was not relevant to flavor, taste, overall acceptability, and purchase intention.

Additionally, in treatments with 1.5% flour, higher pulp content improved taste, consistency, and purchase intention scores compared with average pulp levels. Indeed, HPLF scored higher than MPLF in these sensory attributes. The results indicate that the 10% pulp concentration could have a positive sensory effect on artisanal fermented milk beverages with 1.5% flour. This acceptance could be attributed to the content of ascorbic acid and phenolic compounds that pulp contains. Together with the volatile substances of fermented beverages, they provide better odor and flavor to the product [44]. In terms of purchase intention, the low valuations obtained by the evaluated samples (2.13 to 2.71) were close to the term “maybe buy, maybe not buy”; this could be attributed to the strong flavor that cupuassu pulp has and the lack of familiarity with the consumption of this type of fruit [13,21].

Table 7 shows the values obtained on the JAR scale, used to identify the optimum intensity of each of the attributes of the artisanal fermented milk beverages. The parameters of aroma (acid, alcoholic, cupuassu, and milk), color (white and beige), flavor (caramel), and texture (sandiness, consistency, viscosity, and mouthfeel) showed no significant difference (*p >* 0.05) among the samples. However, pulp concentration was a strong predictor of an alcoholic aroma (R_v_ = 0.837; *p =* 0.034; Appendix A), which, at the same time, had an inverse correlation with the *b** color parameter (yellowness), that is, the higher the perception of *b** was, the lower the alcoholic aroma was (R_v_ = 0.875; *p =* 0.020; Appendix A). In the samples, pH was highly predictive of a cupuassu aroma (R_v_ = 0.985; *p =* 0.007; Appendix A), which was generated by the decrease in pulp and the increase in flour in the beverage (Appendix A). Furthermore, with the concentration of 1.5% flour, a higher percentage of pulp helped to reduce the bitterness (HPLF < LPLF; *p <* 0.05).

Pulp could also contribute to elevate cupuassu flavor, but this was only significant at the concentration of 3% flour (HPHF > LPHF; *p <* 0.05). On the other hand, flour concentration contributed to a less sweet taste but a more acidic and bitter taste. Indeed, the acidic score of MPHF was higher than those of HPLF and LPLH. In terms of bitter taste, the scores of HPHF and MPHF were superior to that of HPLF. In contrast, MPHF and LPHF had sweet taste scores that were lower than those of LPLF and HPLF. This pattern was attributed to the correlation between flour concentration and bitter taste, where flour addition was a strong predictor of bitter taste (R_v_ = 0.928; *p =* 0.006; Appendix A), which was also positively affected by WHC and reduced syneresis (Appendix A).

Flour also favored the darkening of the beverages. Flour increased brown color perception scores and decreased white color perception scores (*p =* 0.002; Appendix A), such that all treatments with 3% flour had higher browning scores than MPLF and LPLF, which was attributed to decreased syneresis and increased WHC (Appendix A–D). However, flour did not influence the score obtained in cupuassu flavor (*p >* 0.05), unlike the pulp percentages, which generated increases in *L** and *C** color parameters, favoring cupuassu flavor (*p =* 0.036; Appendix A) with the increase in pulp percentage (Appendix A).

In terms of texture, the firmness attribute was the one that showed variations among samples (*p <* 0.05) but no correlation concerning pulp and flour concentration (Appendix A), in contrast to the mouthfeel attribute, which was favored by apparent viscosity, resulting from consistency index K, and the addition was inversely related to the behavior of *a** (Appendix A–C). The results indicate that 3% cupuassu flour could increase the acid and bitter taste, brown color, and firm texture of the artisanal fermented milk beverages. Likewise, adding 1.5% cupuassu flour increased the sweet taste values.

According to the results, a penalty analysis of the JAR scores was conducted (Table 8) to improve the formulations of the artisanal fermented milk beverage by adding cupuassu pulp and flour. Parameters with penalty score > 0.5 and incidence > 20% were considered attributes detrimental to overall acceptability. All treatments were penalized for excessive acid aroma, acid taste, and bitter taste, as well as for lack of milk aroma, sweet taste, caramel taste, consistency, firmness, viscosity, and mouthfeel.

Samples HPHF, MPHF, HPLF, and MPLF with 10% or 7.5% pulp were penalized for an excessive alcoholic aroma, which may be because cupuassu pulp represents an important source of substrates for the group of microorganisms (yeasts, lactic, and acetic acid bacteria) of the cupuassu microbiota, which are involved in alcoholic fermentation [47]. Accordingly, MPHF was penalized for an excessive cupuassu aroma, while LPHF and LPLF were penalized for the lack of this aroma. Similarly, HPHF, MPHF, and MPLF presented excessive cupuassu flavor, contrary to LPHF (5% pulp and 3% flour), which was penalized for lack of flavor. This indicates that the ideal pulp concentration has yet to be achieved. Samples with 1.5% flour (HPLF, LPLF, and MPLF) were penalized for lacking brown color. For the brown color attribute, 3% flour was a more promising concentration.

However, it is important to note that a higher flour concentration increased acidity and bitterness, which were penalized attributes. Thus, an ideal flour concentration to harmonize these effects must be established. LPHF, MPHF, and MPLF presented excessive sandiness. Despite the penalty results, most evaluated attributes were close to “moderately more than ideal” (JAR between 2.49 and 3.85). The exceptions were sweetness and caramel flavor, which were less than “ideal” (JAR between 1.74 and 2.28) in all treatments. This can be attributed to the low percentage of sugar in the beverage (3%). The combination of the JAR profile data and the penalty analysis made it possible to establish an ideal product with an adjustment of the percentages of cupuassu pulp and flour, increasing the percentage of sweetener in the product, which would also reduce the acid aroma, acid taste, and bitter taste.

Figure 8 shows the projection of the CATA data of the artisanal fermented milk beverage samples. It explains 94.59% of the total variation of the data, with F1 of 87.06% and F2 of 7.53%. According to the values obtained in the Cochran’s Q-test, seven terms were statistically different among the artisanal fermented milk beverages (*p <* 0.05). These terms were homogeneous appearance, dark brown color, light brown, beige color, white color, bitter taste, and cocoa aroma. Samples containing 3% cupuassu flour (HPHF, MPHF, and LPHF) were described as having bitter taste, dark brown color, and homogeneous appearance. MPHF and LPHF were also associated with a cocoa aroma, which could be attributed to the aromatic compounds in the fermented cupuassu and cocoa seeds, which impart an identical aroma [18].

On the other hand, samples containing 1.5% cupuassu flour (HPLF, MPLF, and LPLF) were described as white, light brown, and beige. These results showed similar behavior in the evaluators’ scores, for both CATA and JAR, where the samples with the highest concentration of cupuassu flour (3%) were perceived as having the most bitter taste and brown color. Likewise, the white and beige colors were associated with the samples with the lowest concentration of cupuassu flour (1.5%).

## 4. Conclusions

Cupuassu pulp and flour had positive effects on some physicochemical parameters of artisanal fermented milk beverages by reducing syneresis and increasing water retention capacity. The high pulp concentration (10%) improved the rheological parameters, especially the consistency index and apparent viscosity. In addition, flour increased protein and fiber contents in the beverages. Likewise, these two ingredients affected the instrumental color parameters, where flour decreased the values of *L**, *b**, and *C**, and pulp increased the values of *a** due to the pigmentation that each one presented as a result of the technological process. Therefore, the combination of samples in the sensory analysis allowed us to observe the effect that flour generated on the product, compared with the concentrations of cupuassu pulp studied by other researchers, where the pigmentation of flour allowed us to improve the color parameter in sensory acceptability. This study established that a higher flour concentration improves sensory appearance, consistency, and firmness in combined formulations, especially when containing 10% pulp. Pulp provided an alcoholic aroma and a cupuassu flavor. At the same time, flour contributed to an excessively acidic and bitter taste. Despite improving the rheological parameters by adding cupuassu, the beverages were still sensorially penalized regarding suboptimal firmness, consistency, and viscosity. Therefore, elaborating an artisanal fermented milk beverage with functional potential using cupuassu pulp or seed flour as ingredients is promising due to physicochemical, nutritional, and microbial quality. However, it is recommended, for future research, to optimize beverage formulations, mainly regarding the concentrations of cupuassu ingredients (e.g., using other percentages of flour and sugar), and even the forms of addition, to improve the characteristics and achieve sensory fullness.

## Figures and Tables

**Figure 2 foods-12-02217-f002:**
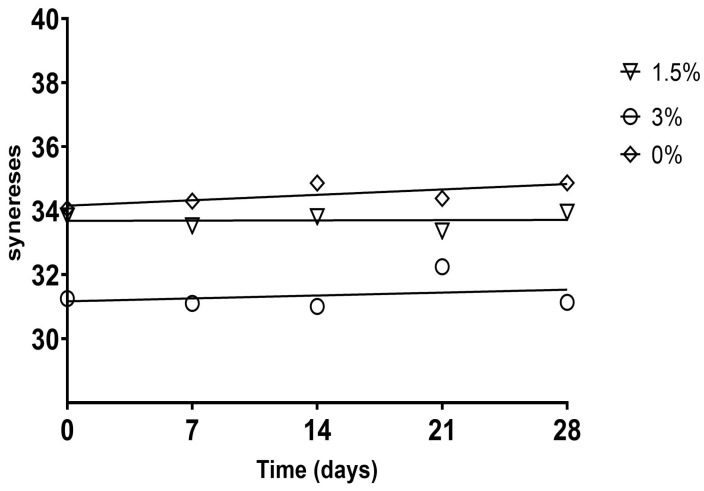
Significant correlations (*p <* 0.05) internally validated with the bootstrap method between syneresis and cupuassu flour in artisanal fermented milk beverages stored at 4 °C.

**Figure 3 foods-12-02217-f003:**
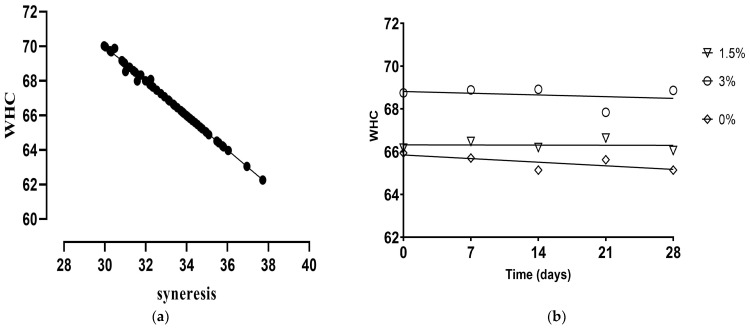
Significant correlations (*p <* 0.05) internally validated with the bootstrap method (**a**) between WHC and syneresis and (**b**) between WHC and flour in artisanal fermented milk beverages stored at 4 °C.

**Figure 4 foods-12-02217-f004:**
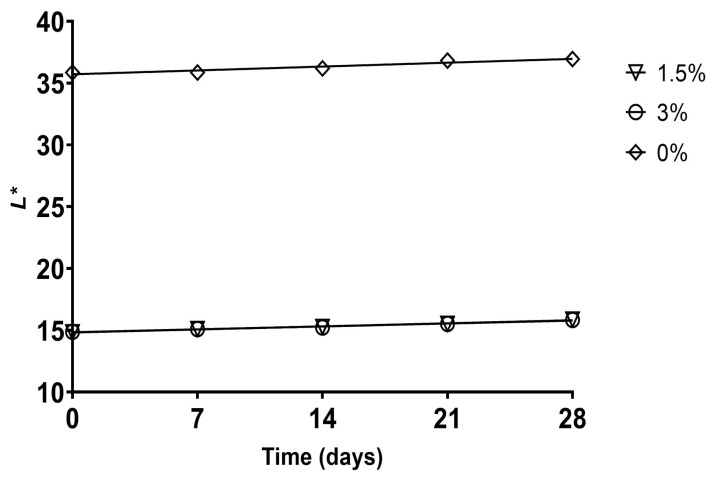
Significant correlations (*p <* 0.05) internally validated with the bootstrap method between *L** and cupuassu flour in artisanal fermented milk beverages stored at 4 °C.

**Figure 5 foods-12-02217-f005:**
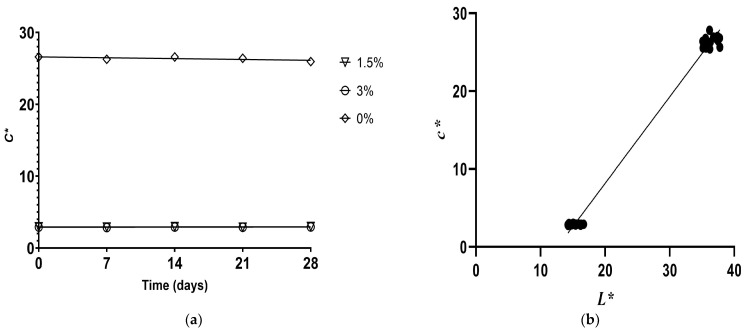
Significant correlations (*p <* 0.05) internally validated with the bootstrap method (**a**) between *C** and cupuassu flour and (**b**) between *C** and *L** in artisanal fermented milk beverages stored at 4 °C.

**Figure 6 foods-12-02217-f006:**
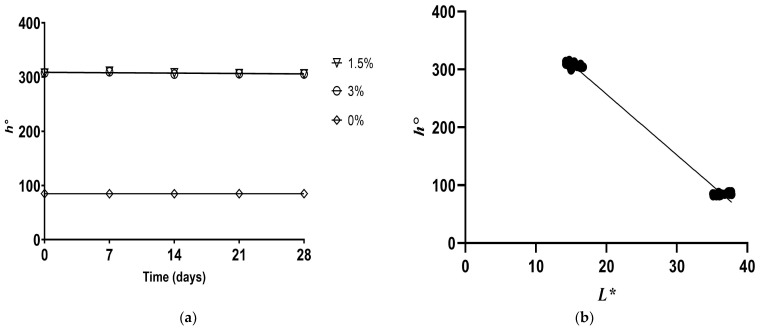
Significant correlations (*p <* 0.05) internally validated with the bootstrap method (**a**) between *h°* and cupuassu flour and (**b**) between *h°* and *L** in artisanal fermented milk beverages stored at 4 °C.

**Figure 8 foods-12-02217-f008:**
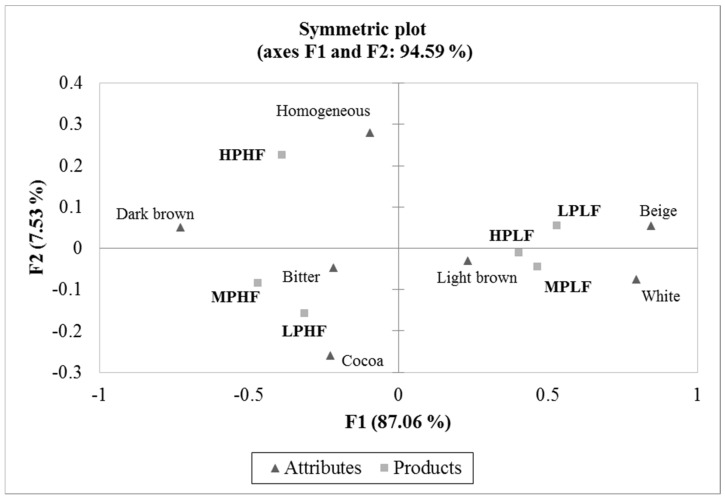
Correspondence analysis of the CATA attribute terms for artisanal fermented milk beverages with cupuassu pulp and flour.

**Table 1 foods-12-02217-t001:** Proximate composition of the freshly prepared artisanal fermented milk beverages.

Sample	Moisture (%)	Ash (%)	Protein (%)	Fat (%)	Fiber (%)	CHOS (%)
NPNF	88.76 ± 0.19 ^c^	0.60 ± 0.01 ^i^	1.58 ± 0.02 ^f^	2.28 ± 0.20 ^b^	0.07 ± 0.03 ^d^	6.71 ± 0.03 ^c^
NPLF	88.22 ± 0.10 ^cd^	0.73 ± 0.01 ^f^	1.92 ± 0.03 ^c^	1.27 ± 0.06 ^de^	0.29 ± 0.17 ^abcd^	7.56 ± 0.11 ^b^
NPHF	84.64 ± 0.14 ^h^	0.78 ± 0.01 ^de^	4.42 ± 0.04 ^a^	2.38 ± 0.13 ^b^	0.39 ± 0.20 ^abc^	7.39 ± 0.23 ^bc^
LPNF	91.63 ± 0.38 ^b^	0.68 ± 0.01 ^g^	1.33 ± 0.01 ^g^	1.08 ± 0.02 ^ef^	0.11 ± 0.01 ^cd^	5.18 ± 0.39 ^de^
MPNF	92.16 ± 0.17 ^b^	0.64 ± 0.01 ^h^	0.84 ± 0.01 ^i^	0.69 ± 0.01 ^g^	0.14 ± 0.02 ^bcd^	5.53 ± 0.18 ^d^
HPNF	93.02 ± 0.20 ^a^	0.63 ± 0.01 ^h^	0.94 ± 0.01 ^h^	0.80 ± 0.01 ^fg^	0.12 ± 0.01 ^cd^	4.48 ± 0.18 ^e^
HPLF	87.16 ± 0.07 ^e^	0.80 ± 0.02 ^d^	1.72 ± 0.05 ^de^	1.87 ± 0.22 ^c^	0.42 ± 0.04 ^ab^	8.03 ± 0.11 ^b^
MPLF	86.46 ± 0.28 ^f^	0.82 ± 0.01 ^c^	1.71 ± 0.03 ^de^	1.28 ± 0.01 ^de^	0.43 ± 0.06 ^ab^	9.30 ± 0.31 ^a^
LPLF	86.03 ± 0.47 ^fg^	0.76 ± 0.01 ^e^	1.64 ± 0.03 ^ef^	1.59 ± 0.09 ^cd^	0.31 ± 0.11 ^abcd^	9.67 ± 0.48 ^a^
HPHF	87.74 ± 0.08 ^de^	0.90 ± 0.01 ^a^	1.86 ± 0.01 ^c^	1.59 ± 0.01 ^cd^	0.56 ± 0.12 ^a^	7.35 ± 0.22 ^bc^
MPHF	85.71 ± 0.10 ^g^	0.86 ± 0.02 ^b^	1.77 ± 0.04 ^d^	1.77 ± 0.17 ^c^	0.52 ± 0.14 ^a^	9.37 ± 0.14 ^a^
LPHF	83.88 ± 0.13 ^i^	0.85 ± 0.01 ^b^	2.38 ± 0.03 ^b^	2.96 ± 0.01 ^a^	0.58 ± 0.03 ^a^	9.35 ± 0.10 ^a^

Results express the means ± SDs of three independent experiments. ^a–i^ Different lowercase superscripts in the same column indicate significant differences among treatments of fermented milk beverage (*p <* 0.05). NPNF, control; NPLF, beverage (1.5% flour); NPHF, beverage (3% flour); LPNF, beverage (5% pulp); MPNF, beverage (7.5% pulp); HPNF, beverage (10% pulp); HPLF, beverage (10% pulp and 1.5% flour); MPLF, beverage (7.5% pulp and 1.5% flour); LPLF, beverage (5% pulp and 1.5% flour); HPHF, beverage (10% pulp and 3% flour); MPHF, beverage (7.5% pulp and 3% flour); LPHF, beverage (5% pulp and 3% flour); CHOS, carbohydrates.

**Table 2 foods-12-02217-t002:** pH of artisanal fermented milk beverages with cupuassu pulp and flour during 28 days of storage at 4 °C.

Parameter	Sample	Storage Time (Days)				
		0	7	14	21	28
**pH**	NPNF	4.40 ± 0.02 ^cA^	4.37 ± 0.01 ^cB^	4.38 ± 0.01 ^dAB^	4.30 ± 0.01 ^dC^	4.37 ± 0.01 ^eB^
	NPLF	4.49 ± 0.03 ^bA^	4.48 ± 0.01 ^abA^	4.50 ± 0.01 ^bA^	4.48 ± 0.02 ^bA^	4.47 ± 0.01 ^aC^
	NPHF	4.67 ± 0.02 ^aA^	4.53 ± 0.01 ^aD^	4.63 ± 0.01 ^aAB^	4.59 ± 0.01 ^aBC^	4.58 ± 0.03 ^aC^
	LPNF	4.18 ± 0.03 ^gB^	4.13 ± 0.01 ^fC^	4.19 ± 0.01 ^hAB^	4.14 ± 0.02 ^gC^	4.23 ± 0.01 ^gA^
	MPNF	4.11 ± 0.02 ^hBC^	4.19 ± 0.04 ^eA^	4.10 ± 0.01 ^jBC^	4.09 ± 0.01 ^hC^	4.14 ± 0.01 ^hAB^
	HPNF	3.99 ± 0.02 ^iC^	4.11 ± 0.03 ^fAB^	4.14 ± 0.010 ^iA^	3.98 ± 0.01 ^iC^	4.08 ± 0.01 ^iB^
	HPLF	4.21 ± 0.02 ^fgA^	4.21 ± 0.01 ^eA^	4.24 ± 0.01 ^gA^	4.10 ± 0.01 ^hB^	4.12 ± 0.03 ^hiB^
	MPLF	4.23 ± 0.01 ^efgAB^	4.23 ± 0.02 ^eAB^	4.22 ± 0.01 ^gAB^	4.25 ± 0.02 ^eA^	4.21 ± 0.02 ^gB^
	LPLF	4.28 ± 0.01 ^deBC^	4.29 ± 0.01 ^dAB^	4.30 ± 0.01 ^fAB^	4.26 ± 0.01 ^eC^	4.31 ± 0.01 ^fA^
	HPHF	4.30 ± 0.04 ^dA^	4.30 ± 0.02 ^dA^	4.30 ± 0.01 ^fA^	4.19 ± 0.01 ^fC^	4.34 ± 0.01 ^efA^
	MPHF	4.25 ± 0.01 ^defC^	4.34 ± 0.01 ^cdB^	4.35 ± 0.01 ^eB^	4.25 ± 0.01 ^eC^	4.42 ± 0.01 ^dA^
	LPHF	4.44 ± 0.01 ^bcB^	4.43 ± 0.03 ^bB^	4.45 ± 0.01 ^cB^	4.38 ± 0.01 ^cC^	4.52 ± 0.01 ^bA^

Results express the means ± SDs of three independent experiments. ^a–i^ Different lowercase superscripts in the same column indicate significant differences among treatments of artisanal fermented milk beverage (*p <* 0.05). ^A–D^ Different uppercase superscripts in the same row indicate significant differences among storage times (*p <* 0.05). NPNF, control; NPLF, beverage (1.5% flour); NPHF, beverage (3% flour); LPNF, beverage (5% pulp); MPNF, beverage (7.5% pulp); HPNF, beverage (10% pulp); HPLF, beverage (10% pulp and 1.5% flour); MPLF, beverage (7.5% pulp and 1.5% flour); LPLF, beverage (5% pulp and 1.5% flour); HPHF, beverage (10% pulp and 3% flour); MPHF, beverage (7.5% pulp and 3% flour); LPHF, beverage (5% pulp and 3% flour).

**Table 3 foods-12-02217-t003:** Syneresis and water-holding capacity of artisanal fermented milk beverages with cupuassu pulp and flour during 28 days of storage at 4 °C.

Parameter	Sample	Storage Time (Days)				
		0	7	14	21	28
**Syneresis**	NPNF	36.95 ± 1.0 ^aA^	35.81 ± 2.15 ^aA^	37.73 ± 0.87 ^aA^	35.58 ± 0.72 ^abA^	36.04 ± 0.80 ^aA^
	NPLF	34.77 ± 0.33 ^abcAB^	34.07 ± 0.21 ^abcdB^	35.76 ± 0.79 ^abA^	35.77 ± 0.12 ^aA^	34.29 ± 0.88 ^abAB^
	NPHF	33.79 ± 0.02 ^bcdA^	32.00 ± 0.71 ^defAB^	31.20 ± 1.04 ^fgB^	32.54 ± 0.55 ^deAB^	31.40 ± 1.21 ^cdB^
	LPNF	33.13 ± 0.46 ^bcdeC^	34.97 ± 0.36 ^abAB^	35.48 ± 0.12 ^bcA^	33.85 ± 1.03 ^bcdBC^	34.17 ± 0.26 ^bABC^
	MPNF	34.67 ± 0.16 ^abcA^	34.42 ± 0.39 ^abcA^	33.99 ± 1.26 ^bcdA^	34.68 ± 0.58 ^abcA^	34.96 ± 0.03 ^abA^
	HPNF	31.49 ± 0.20 ^defC^	31.99 ± 0.88 ^defBC^	32.24 ± 0.50 ^defBC^	33.40 ± 0.01 ^cdAB^	34.29 ± 0.57 ^abA^
	HPLF	32.36 ± 0.05 ^cdefB^	32.74 ± 0.05 ^bcdeB^	32.20 ± 0.10 ^defB^	32.92 ± 0.65 ^dB^	33.81 ± 0.16 ^bA^
	MPLF	33.11 ± 1.0 ^bcdeAB^	32.34 ± 0.20 ^cdefB^	33.55 ± 0.06 ^cdeAB^	33.87 ± 0.54 ^bcdA^	34.53 ± 0.45 ^abA^
	LPLF	35.09 ± 2.34 ^abA^	34.94 ± 0.34 ^abA^	33.72 ± 0.74 ^bcdeBC^	30.89 ± 1.01 ^eB^	33.16 ± 0.04 ^bcBC^
	HPHF	29.97 ± 0.16 ^fA^	30.32 ± 0.38 ^fA^	30.03 ± 0.29 ^gA^	30.83 ± 0.50 ^eA^	30.25 ± 0.74 ^dA^
	MPHF	30.97 ± 0.65 ^efA^	30.48 ± 0.93 ^efA^	31.01 ± 0.99 ^fgA^	32.25 ± 0.01 ^deA^	31.67 ± 0.45 ^cdA^
	LPHF	30.28 ± 0.52 ^fC^	31.59 ± 0.76 ^efBC^	31.77 ± 0.21 ^efgB^	33.35 ± 0.04 ^cdA^	31.22 ± 0.61 ^dBC^
**WHC**	NPNF	63.05 ± 1.00 ^fA^	64.18 ± 2.15 ^eA^	62.26 ± 0.87 ^gA^	64.42 ± 0.72 ^efA^	63.97 ± 0.80 ^dA^
	NPLF	65.23 ± 0.33 ^defAB^	65.93 ± 0.21 ^bcdeA^	64.24 ± 0.79 ^fB^	64.23 ± 0.12 ^fB^	65.71 ± 0.88 ^cdAB^
	NPHF	66.21 ± 0.02 ^cdeB^	68.00 ± 0.71 ^abAB^	68.80 ± 1.04 ^abA^	67.46 ± 0.55 ^abcAB^	68.60 ± 1.21 ^abA^
	LPNF	66.87 ± 0.46 ^bcdeA^	65.03 ± 0.36 ^deBC^	64.52 ± 0.12 ^efC^	66.14 ± 1.03 ^cdeAB^	65.83 ± 0.26 ^cABC^
	MPNF	65.33 ± 0.16 ^defA^	65.58 ± 0.39 ^cdeA^	66.00 ± 1.26 ^defA^	65.32 ± 0.58 ^defA^	65.04 ± 0.03 ^cdA^
	HPNF	68.51 ± 0.20 ^abcA^	68.01 ± 0.88 ^abAB^	67.76 ± 0.50 ^bcdAB^	66.60 ± 0.01 ^bcdBC^	65.71 ± 0.57 ^cdC^
	HPLF	67.64 ± 0.05 ^abcdA^	67.26 ± 0.06 ^bcdA^	67.80 ± 0.10 ^bcdA^	67.08 ± 0.65 ^bcdA^	66.19 ± 0.16 ^cB^
	MPLF	66.89 ± 1.01 ^bcdeAB^	67.66 ± 0.20 ^abcA^	66.45 ± 0.06 ^cdeAB^	66.13 ± 0.54 ^cdeB^	65.47 ± 0.45 ^cdB^
	LPLF	64.88 ± 2.34 ^efB^	65.06 ± 0.34 ^deB^	66.28 ± 0.74 ^deAB^	69.11 ± 1.01 ^aA^	66.84 ± 0.04 ^bcAB^
	HPHF	70.03 ± 0.16 ^aA^	69.68 ± 0.38 ^aA^	69.97 ± 0.29 ^aA^	69.17 ± 0.50 ^aA^	69.75 ± 0.74 ^aA^
	MPHF	69.03 ± 0.65 ^abAB^	69.88 ± 0.69 ^aA^	68.54 ± 0.61 ^abAB^	68.08 ± 0.58 ^abB^	68.33 ± 0.45 ^abAB^
	LPHF	69.71 ± 0.56 ^aA^	67.99 ± 0.26 ^abB^	68.34 ± 0.07 ^abcB^	66.65 ± 0.04 ^bcdC^	68.78 ± 0.61 ^aAB^

Results express the means ± SDs of three independent experiments. ^a–g^ Different lowercase superscripts in the same column indicate significant differences among treatments of artisanal fermented milk beverage (*p <* 0.05). ^A–C^ Different uppercase superscripts in the same row indicate significant differences among storage times (*p <* 0.05). NPNF, control; NPLF, beverage (1.5% flour); NPHF, beverage (3% flour); LPNF, beverage (5% pulp); MPNF, beverage (7.5% pulp); HPNF, beverage (10% pulp); HPLF, beverage (10% pulp and 1.5% flour); MPLF, beverage (7.5% pulp and 1.5% flour); LPLF, beverage (5% pulp and 1.5% flour); HPHF, beverage (10% pulp and 3% flour); MPHF, beverage (7.5% pulp and 3% flour); LPHF, beverage (5% pulp and 3% flour); WHC, water-holding capacity.

**Table 6 foods-12-02217-t006:** Sensory acceptance of artisanal fermented milk beverages with cupuassu pulp and flour.

Sample	Attribute ^1)^							
	Appearance	Color	Flavor	Taste	Consistency	Firmness	Overall Acceptability	Purchase Intention
HPLF	6.03 ± 2.02 ^abc^	5.79 ± 1.99 ^bc^	6.30 ± 1.57 ^a^	5.59 ± 2.09 ^a^	6.07 ± 1.93 ^a^	6.03 ± 1.91 ^ab^	5.49 ± 1.82 ^a^	2.71 ± 1.10 ^a^
MPLF	5.15 ± 2.15 ^c^	5.16 ± 2.06 ^c^	5.77 ± 1.87 ^a^	4.39 ± 2.13 ^b^	5.00 ± 1.97 ^b^	5.08 ± 1.88 ^b^	4.62 ± 1.99 ^a^	2.13 ± 1.02 ^b^
LPLF	5.43 ± 2.11 ^bc^	5.48 ± 2.12 ^c^	5.92 ± 1.70 ^a^	5.23 ± 1.97 ^ab^	5.98 ± 1.81 ^a^	5.70 ± 1.80 ^ab^	5.51 ± 1.77 ^a^	2.59 ± 1.10 ^ab^
HPHF	6.89 ± 1.72 ^a^	6.93 ± 1.63 ^a^	6.48 ± 1.54 ^a^	4.67 ± 2.50 ^ab^	6.18 ± 1.74 ^a^	6.31 ± 1.81 ^a^	5.26 ± 2.26 ^a^	2.33 ± 1.11 ^ab^
MPHF	6.39 ± 1.80 ^ab^	6.56 ± 1.74 ^ab^	6.41 ± 1.90 ^a^	4.59 ± 2.28 ^ab^	5.97 ± 1.83 ^a^	6.02 ± 1.80 ^ab^	5.02 ± 2.12 ^a^	2.23 ± 1.09 ^ab^
LPHF	6.36 ± 1.76 ^ab^	6.59 ± 1.53 ^ab^	6.61 ± 1.45 ^a^	4.72 ± 2.31 ^ab^	6.08 ± 1.78 ^a^	6.31 ± 1.65 ^a^	5.20 ± 2.08 ^a^	2.20 ± 1.03 ^ab^

^a–c^ Different lowercase superscripts indicate significant differences among treatments of artisanal fermented milk beverage (*p <* 0.05). ^1)^ Purchase intention was evaluated on a structured 5-point hedonic scale, whereas the other attributes were evaluated on a 9-point hedonic scale. HPLF, beverage (10% pulp and 1.5% flour); MPLF, beverage (7.5% pulp and 1.5% flour); LPLF, beverage (5% pulp and 1.5% flour); HPHF, beverage (10% pulp and 3% flour); MPHF, beverage (7.5% pulp and 3% flour); LPHF, beverage (5% pulp and 3% flour).

**Table 7 foods-12-02217-t007:** Just-about-right (JAR) profile scores for the different formulations of artisanal fermented milk beverage evaluated.

Sample	Aroma												Taste		
	Acid			Alcoholic			Cupuassu			Milk			Sweet	Acid	Bitter
**HPLF**	3.25 ± 0.62 ^a^			3.21 ± 0.68 ^a^			3.03 ± 0.83 ^a^			2.69 ± 0.78 ^a^			2.28 ± 1.74 ^a^	3.23 ± 0.62 ^b^	3.25 ± 0.71 ^c^
**MPLF**	3.38 ± 0.80 ^a^			3.13 ± 0.79 ^a^			3.03 ± 0.83 ^a^			2.57 ± 0.77 ^a^			1.98 ± 0.84 ^ab^	3.51 ± 0.95 ^ab^	3.46 ± 0.95 ^abc^
**LPLF**	3.08 ± 0.69 ^a^			3.07 ± 0.57 ^a^			2.98 ± 0.85 ^a^			2.57 ± 0.89 ^a^			2.28 ± 0.85 ^a^	3.33 ± 0.63 ^b^	3.33 ± 0.82 ^ab^
**HPHF**	3.26 ± 0.60 ^a^			3.20 ± 0.87 ^a^			3.00 ± 0.91 ^a^			2.54 ± 0.87 ^a^			1.85 ± 0.77 ^ab^	3.39 ± 0.88 ^ab^	3.85 ± 0.79 ^a^
**MPHF**	3.43 ± 0.83 ^a^			3.23 ± 0.91 ^a^			3.10 ± 0.82 ^a^			2.71 ± 0.80 ^a^			1.74 ± 0.72 ^b^	3.79 ± 0.86 ^a^	3.74 ± 0.93 ^ab^
**LPHF**	3.25 ± 0.65 ^a^			3.07 ± 0.71 ^a^			2.72 ± 0.93 ^a^			2.59 ± 0.85 ^a^			1.72 ± 0.76 ^b^	3.39 ± 0.89 ^ab^	3.69 ± 0.99 ^abc^
**Sample**	Color												Flavor		
	White				Brown				Beige				Cupuassu		Caramel
**HPLF**	2.97 ± 0.97 ^a^				2.74 ± 1.03 ^ab^				2.95 ± 0.83 ^a^				3.23 ± 0.71 ^ab^		2.03 ± 0.69 ^a^
**MPLF**	3.03 ± 1.02 ^a^				2.54 ± 0.70 ^b^				2.79 ± 0.78 ^a^				3.16 ± 0.94 ^ab^		2.03 ± 0.94 ^a^
**LPLF**	3.10 ± 1.08 ^a^				2.62 ± 0.67 ^b^				3.01 ± 0.77 ^a^				3.08 ± 0.87 ^ab^		2.00 ± 0.77 ^a^
**HPHF**	2.61 ± 0.94 ^a^				3.08 ± 0.67 ^a^				2.82 ± 0.67 ^a^				3.39 ± 1.16 ^a^		1.98 ± 0.99 ^a^
**MPHF**	2.71 ± 0.94 ^a^				3.08 ± 0.46 ^a^				2.84 ± 0.78 ^a^				3.25 ± 0.99 ^ab^		2.02 ± 0.89 ^a^
**LPHF**	2.69 ± 0.79 ^a^				3.03 ± 0.61 ^a^				2.89 ± 0.64 ^a^				2.85 ± 1.12 ^b^		2.10 ± 0.98 ^a^
**Sample**	Texture														
	Sandiness				Consistency				Firmness				Viscosity		Mouthfeel
**HPLF**	3.08 ± 0.82 ^a^				2.82 ± 0.70 ^a^				2.77 ± 0.59 ^ab^				2.79 ± 0.72 ^a^		2.85 ± 0.76 ^a^
**MPLF**	3.10 ± 0.87 ^a^				2.49 ± 0.72 ^a^				2.51 ± 0.72 ^b^				2.69 ± 0.75 ^a^		2.54 ± 0.98 ^a^
**LPLF**	3.08 ± 0.79 ^a^				2.62 ± 0.64 ^a^				2.57 ± 0.65 ^ab^				2.71 ± 0.72 ^a^		2.61 ± 0.92 ^a^
**HPHF**	3.20 ± 0.68 ^a^				2.75 ± 0.75 ^a^				2.80 ± 0.70 ^ab^				2.85 ± 0.83 ^a^		2.66 ± 1.01 ^a^
**MPHF**	3.08 ± 0.77 ^a^				2.82 ± 0.62 ^a^				2.85 ± 0.66 ^a^				2.71 ± 0.65 ^a^		2.56 ± 0.95 ^a^
**LPHF**	3.20 ± 0.64 ^a^				2.72 ± 0.58 ^a^				2.77 ± 0.58 ^ab^				2.89 ± 0.63 ^a^		2.56 ± 1.00 ^a^

^a–c^ Different lowercase superscripts indicate significant differences among treatments of artisanal fermented milk beverage (*p <* 0.05). HPLF, beverage (10% pulp and 1.5% flour); MPLF, beverage (7.5% pulp and 1.5% flour); LPLF, beverage (5% pulp and 1.5% flour); HPHF, beverage (10% pulp and 3% flour); MPHF, beverage (7.5% pulp and 3% flour); LPHF, beverage (5% pulp and 3% flour).

**Table 8 foods-12-02217-t008:** Consumer penalty analysis of the just-about-right (JAR) diagnostic attributes of artisanal fermented milk beverages.

Attribute		Sample											
		HPHF		LPHF		MPHF		HPLF		LPLF		MPLF	
		Not Enough	Too Much	Not Enough	Too Much	Not Enough	Too Much	Not Enough	Too Much	Not Enough	Too Much	Not Enough	Too Much
**Aroma**	Acid	- ^1^	24.59% (0.81) ^2^	-	29.51% (1.68)	-	37.70% (1.01)	-	29.51% (1.67)	-	21.31% (1.11)	-	34.43% (0.81)
	Alcoholic	-	31.15% (0.83)	-	-	-	31.15% (1.01)	-	27.87% (0.84)	-	-	-	24.59% (1.60)
	Cupuassu	26.23% (1.05)		37.70% (1.23)	-	-	22.95% (0.75)	-	-	27.87% (1.39)	-	-	-
	Milk	47.54% (0.9)	-	39.34% (1.10)	-	34.43% (1.33)	-	31.15% (1.34)	-	40.98% (1.33)	-	39.34% (0.70)	-
**Taste**	Sweet	80.33% (2.74)	-	81.97% (2.53)	-	83.61% (2.37)	-	68.85% (1.81)	-	62.30% (1.47)	-	73.77% (0.89)	-
	Acid	-	44.26% (1.03)	-	39.34% (1.92)	-	57.38% (1.46)	-	27.87% (1.73)	-	36.07% (1.46)	-	44.26% (1.40)
	Bitter	-	67.21% (1.60)	-	54.10% (2.42)	-	60.66% (1.18)	-	29.51% (0.95)	-	34.43% (1.56)	-	45.90% (0.29)
**Color**	White	-	-	-	-	-	-		-	-	-	-	-
	Brown	-	-	-	-	-	-	37.70% (1.21)	-	37.70% (1.02)	-	44.26% (1.23)	-
	Beige	-	-	-	-	-	-	-	-	-	-	-	-
**Flavor**	Cupuassu	-	44.26% (1.03)	36.07% (1.55)	-	-	36.07% (1.51)	-	-	-	26.23% (1.32)	-	31.15% (0.90)
	Caramel	75.41% (1.93)	-	70.49% (1.14)	-	75.41% (1.87)	-	75.41% (0.85)	-	77.05% (2.17)	-	75.41% (1.01)	-
**Texture**	Sandiness	-	-	-	24.59% (1.08)	-	-	-	22.95% (0.54)	-	-	-	24.59% (1.61)
	Consistency	27.87% (1.77)	-	27.87% (1.56)	-	21.31% (0.12)	-	27.87% (1.91)	-	39.34% (1.90)	-	49.18% (0.93)	-
	Firmness	21.31% (2.16)	-	27.87% (1.43)	-	-	-	27.87% (2.21)	-	40.98% (1.97)	-	47.54% (1.21)	-
	Viscosity	26.23% (1.90)	-	21.31% (0.91)	-	29.51% (0.94)	-	27.87% (1.39)	-	31.15% (1.39)	-	36.07% (1.46)	-
	Mouthfeel	45.90% (1.61)	-	40.98% (1.71)	-	44.26% (2.00)	-	29.51% (1.72)	-	39.34% (1.76)	-	49.18% (1.81)	-

^1^ (-) indicates that less than 20% of consumers chose that JAR category. ^2^ Percentage of consumers and mean decreases. HPHF, beverage (10% pulp and 3% flour); LPHF, beverage (5% pulp and 3% flour); MPHF, beverage (7.5% pulp and 3% flour); HPLF, beverage (10% pulp and 1.5% flour); LPLF, beverage (5% pulp and 1.5% flour); MPLF, beverage (7.5% pulp and 1.5% flour).

## Data Availability

The datasets generated for this study are available on request to the corresponding author.

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
