# Peer review of "Physicochemical, Rheological, and Nutritional Quality of Artisanal Fermented Milk Beverages with Cupuassu (Theobroma grandiflorum) Pulp and Flour"

_foods, 2023, doi:10.3390/foods12112217_

Round 1
Reviewer 1 Report
REVIEWER COMMENTS
Manuscript Title: Cupuassu (Theobroma grandiflorum) pulp and flour improve physicochemical, rheological, and nutritional quality of artisanal fermented milk beverages
This research paper presented for review is interesting for science, producer and consumers. The study is well-designed, the text clear and easy to read. However, there are some important insufficients
I have some comments to the authors as follows:
1. Title could be more effective, for example, delete “improve” and The title may be “Physicochemical, rheological, and nutritional quality of artisanal fermented milk beverages produced with cupuassu (Theobroma grandiflorum) pulp and flour”
2. L 41: Keywords may be: cupuassu; Theobroma grandiflorum; milk beverage; by-products; color analysis; rheological behavior; sensory acceptance
3. L 100-101: Why didn't you prepare Cupuassu (Theobroma grandiflorum) pulp and flour yourself?, why didn't you determine its properties yourself? If these are already ready-made products, are they standard features? and what are they used for? Please add a detailed explanation about sample supply and features to the manuscript.
4. L 56-57: please review the whey knowledge with other references and revise the sentence (which proteins, lipids etc. and whey represents up to 95% of the volume of milk..?)
5. L 99: What were the properties of whey?
6. L 104: “…pasteurized whey (49%) and milk (51%)..” Why was this ratio not equal?
7. L 107: “bulgaricus” instead of “.. Bulgaricus..”
8. L575….: It could not be understood that the samples selected for sensory analysis were selected based on the results of previous (2015-2017) research. As a result, another product was studied in this research and why were 12 samples made?
9. L: 125 not “0”, please change with “1” (Abstract, all manuscript, tables, figures)
10. L: 129-130 Please write your analysis results
11. L: 363-364 Figure 2 (y axis: not synereses) syneresis
12. Since sensory analysis is not performed in all samples, the Conclusions section should be written more appropriately, and/or the sensory characteristics of all samples, if any, should be given in the relevant section.
13. References section should be revised. For example, the titles of the articles are lowercase or uppercase, some journal titles are long/short spelled, etc. There is no standardization in almost all of this section.

Author Response
GENERAL COMMENTS BY THE AUTHORS:
We would like to thank the reviewers for their insight and thoughtful critique of our manuscript. We believe we have fully addressed all of these concerns and comments, which has increased the overall impact of the manuscript.
The modifications were marked using the "Track Changes" function.
RESPONSE TO REVIEWERS:
Review Report (Round 1)
REVIEWER COMMENTS
Manuscript Title: Cupuassu (Theobroma grandiflorum) pulp and flour improve physicochemical, rheological, and nutritional quality of artisanal fermented milk beverages
This research paper presented for review is interesting for science, producer and consumers. The study is well-designed, the text clear and easy to read. However, there are some important insufficients.
Answer: Thank you for your consideration. We revised the entire manuscript to supply the insufficient appointed by reviewer.
I have some comments to the authors as follows:
1. Title could be more effective, for example, delete “improve” and The title may be “Physicochemical, rheological, and nutritional quality of artisanal fermented milk beverages produced with cupuassu (Theobroma grandiflorum) pulp and flour”
Answer: Thank you for your consideration. The title was modified as suggested.
2. L 41: Keywords may be: cupuassu; Theobroma grandiflorum; milk beverage; by-products; color analysis; rheological behavior; sensory acceptance
Answer: Thank you for your consideration. The keywords were modified as suggested.
3. L 100-101: Why didn't you prepare Cupuassu (Theobroma grandiflorum) pulp and flour yourself?, why didn't you determine its properties yourself? If these are already ready-made products, are they standard features? and what are they used for? Please add a detailed explanation about sample supply and features to the manuscript.
Answer: The pulp and flour were not prepared because they are products widely sold in Brazil, mainly in the Northeast region. Yes, they are a ready-made product, and they also are standard features. Cupuassu pulp and flour are commonly used in the formulation of juice, dessert, fermented milk, dulce de leche, and other typical foods. Thank you for your consideration, more information was added to manuscript.
4. L 59-64: please review the whey knowledge with other references and revise the sentence (which proteins, lipids etc. and whey represents up to 95% of the volume of milk..?)
Answer: Thank you for your consideration. L 60-65 were modified as suggested: “Indeed, whey represents up to 95% of the volume of milk and has high nutritional val-ue, containing 70-72% lactose; 12-15% water-soluble vitamins B (B1, B2, B3, B5, B6, B9, and B12), C and minerals (calcium, potassium, sodium, and magnesium); 8-10 % proteins (α-lactalbumin, β-lactoglobulin, serum bovine albumin, glycomacropeptide, immunoglobulins, lactoperoxidase, and lactoferrin); protein-derived peptides (β-Lactophorin, β-Lactotensin, α-Lactophorin, Albutensin Serophorin, Lactoferricin) and lipids”.
5. L 99: What were the properties of whey?
Answer: Thank you for your consideration. L 108 was modified as suggested: “The sweet whey (pH 6.29 ± 0.02) was acquired from manufacturing fresh cheese made for other research”
6. L 104: “…pasteurized whey (49%) and milk (51%)..” Why was this ratio not equal?
Answer: Ratios of whey and milk were determined according to the Brazilian regulation cited in L56, indicating that fermented milk beverages should contain a minimum concentration of milk base of 51 % vol/vol.
7. bulgaricus L 107: “bulgaricus” instead of “.. Bulgaricus..”
Answer: Thank you for your consideration. L 116 was modified as suggested.
8. L575….: It could not be understood that the samples selected for sensory analysis were selected based on the results of previous (2015-2017) research. As a result, another product was studied in this research and why were 12 samples made?
Answer: The 12 samples were prepared with the aim of observing the physicochemical behavior that could be generated by the addition of cupuassu flour at concentrations (0, 1.5, and 3%), in combination with the addition of pulp (0, 5, 7.5, and 10). However, the number of samples was too high for sensory analysis, which would be too exhaustive for the evaluator. Therefore, a sensory preference test was performed first, in which the samples combined with cupuassu pulp and flour obtained the most favorable result (data not shown). Therefore, the combined samples containing pulp and flour at the concentrations studied were used for the analysis of sensory acceptability, purchase intention, JAR, and CATA. The addition of flour provides expressive modifications in the sensory characteristics, so the sensory evaluation so the sensory evaluation was focused on the combined treatments with flour.
9. L: 125 not “0”, please change with “1” (Abstract, all manuscript, tables, figures)
Answer: The authors consider that the proposed correction should not be made because the first day of analysis was before 24 hours of storage. This day 0 of storage is generally used as a starting point to analyze samples during storage according to the articles mentioned below:
Costa, M.P.; Rosario, A.I.L.; Silva, V.L.; Vieira, C.P.; Conte-Junior, C. A. Rheological, Physical and Sensory Evaluation of Low-Fat Cupuassu Goat Milk Yogurts Supplemented with Fat Replacer. Food Sci Anim Resour 2022, 42, 210-224. https://doi.org/10.5851/kosfa.2021.e64
Jovanović, M.; Zlatanović, S.; Micić, D.; Bacić, D.; Mitić-Ćulafić, D.; Đuriš, M.; Gorjanović, S. Functionality and Palatability of Yogurt Produced Using Beetroot Pomace Flour Granulated with Lactic Acid Bacteria. Foods 2021, 10, 1696. https://doi.org/10.3390/foods10081696
Kaur Sidhu, M.; Lyu, F.; Sharkie, T.P.; Ajlouni, S.; Ranadheera, C.S. Probiotic Yogurt Fortified with Chickpea Flour: Physico-Chemical Properties and Probiotic Survival during Storage and Simulated Gastrointestinal Transit. Foods 2020, 9, 1144. https://doi.org/10.3390/foods9091144
Pereira, A.L.F.; Feitosa, W.S.C.; Abreu, V.K.G.; de Oliveira Lemos, T.; Gomes, W.F.; Narain, N.; Rodrigues, S. Impact of fermentation conditions on the quality and sensory properties of a probiotic cupuassu (Theobroma grandiflorum) beverage. Food Res Int 2017, 100, 603-611. https://doi.org/10.1016/j.foodres.2017.07.055
10. L: 129-130 Please write your analysis results
Answer: The results are analyzed in the session corresponding to the proximal composition in L284-L287.
11. L: 363-364 Figure 2 (y axis: not synereses) syneresis
Answer: Thank you for your consideration. Figure 2 was modified as suggested.
12. Since sensory analysis is not performed in all samples, the Conclusions section should be written more appropriately, and/or the sensory characteristics of all samples, if any, should be given in the relevant section.
Answer: Thank you for your consideration. L 724-729 were modified as suggested: “Therefore, the combination of samples for the sensory analysis allowed to observe the effect that the flour generated in the product, compared to the concentrations of cupuassu pulp studied by other researchers, where the pigmentation of the flour allowed to improve the color parameter in the sensory acceptability”.
13. References section should be revised. For example, the titles of the articles are lowercase or uppercase, some journal titles are long/short spelled, etc. There is no standardization in almost all of this section.
Answer: Thank you for your consideration. The references were modified as suggested.

Reviewer 2 Report
The research is very well thought out and the paper is very well written, I enjoyed reading it. Here are some suggestions that would further improve the manuscript: L 95 Authors should state that the fermented beverage is made from milk and whey (ie., a beverage containing fermented milk and whey) L107 Bulgaricus should be written as bulgaricus L310 The authors should consider presenting the values for day 0 and day 28 in the text, in parentheses, to make the discussion easier to follow. Where the authors refer to Fig. 1 in the text? L 558 The presence of yeast, their number, can vary in pulp and flour, and that can affect the taste of the product. Are the authors thinking of introducing sterilization of flour and pulp in the next research so that the taste will be the same every time? What sterilization methods would be used?
Author Response
GENERAL COMMENTS BY THE AUTHORS:
We would like to thank the reviewers for their insight and thoughtful critique of our manuscript. We believe we have fully addressed all of these concerns and comments, which has increased the overall impact of the manuscript.
The modifications were marked using the "Track Changes" function.
RESPONSE TO REVIEWERS:
Review Report (Round 2)
Comments and Suggestions for Authors
The research is very well thought out and the paper is very well written, I enjoyed reading it. Here are some suggestions that would further improve the manuscript:
Answer: Thank you for your consideration. We revised the entire manuscript to supply the insufficient appointed by reviewer.
L 95 Authors should state that the fermented beverage is made from milk and whey (ie., a beverage containing fermented milk and whey)
Answer: Thank you for your consideration. L 103 was modified as suggested: “fermented milk beverage prepared with milk and whey”
L107 Bulgaricus should be written as bulgaricus
Answer: Thank you for your consideration. L 118 was modified as suggested.
L310 The authors should consider presenting the values for day 0 and day 28 in the text, in parentheses, to make the discussion easier to follow. Where the authors refer to Fig. 1 in the text?
Answer: Thank you for your consideration. L 323 was modified as suggested: “During the storage period, NPNF and the samples with the addition of cupuassu flour showed a decrease in pH values, except for the samples MPHF and LPHF combined with pulp and 3% cupuassu flour, which showed an increase in pH values be-tween day 0 (4.25 and 4.44) and day 28 (4.42 and 4.52, respectively)”. Figure 1 added in text L294.
L 558 The presence of yeast, their number, can vary in pulp and flour, and that can affect the taste of the product. Are the authors thinking of introducing sterilization of flour and pulp in the next research so that the taste will be the same every time? What sterilization methods would be used?
Answer: Thinking from the food and nutritional point of view, we consider that it is not convenient to implement the sterilization of flour and pulp, because the yeasts present in cupuassu have probiotic potential, which are the Pichia yeasts (mentioned in L342), which in addition to being beneficial to the human body, allow improving the physicochemical and sensory characteristics of the product through the fermentation process. Therefore, no sterilization method would be used for this type of product, since the aim is to improve and maintain the functional characteristics of this food.

Round 2
Reviewer 1 Report
Dear author/authors,
The whey-related question and my recommendation didn't convince enough in this revision. should be reconsidered.
Day 1 is more accurate than day 0. Some examples given by the author/authors (day 0), hundreds can be written for day 1.
Please review the answers written for the first revision.
Author Response
Dear reviewer/reviewers,
The whey-related question and my recommendation didn't convince enough in this revision. should be reconsidered.
Answer: Thank you for your consideration. As suggested, we provide more information about the whey used in this study. L:58-67 “In this context, the addition of cheese whey, a by-product of the cheese industry that is generally discarded, as an ingredient in artisanal fermented milk beverages and other healthy food products due to its nutritional quality, can be an alternative to reduce waste. The whey retains 50-55% of milk solids, 70-72% lactose; 12-15% water-soluble vitamins (B and C); and minerals (calcium, potassium, sodium, and magnesium); 8-10% proteins (α-lactalbumin, β-lactoglobulin, serum bovine albumin, glycomacropeptide, immunoglobulins, lactoperoxidase, and lactoferrin), protein-derived peptides (β-Lactophorin, β-Lactotensin, α-Lactophorin, Albutensin Serophorin, Lactoferricin) and lipids which provide innumerable health benefits”.
L:110-118 “The sweet whey (pH 6.29 ± 0.02) obtained from the processing of fresh cheese and used for this study had the following proximal composition (Figure S1): dry matter (7.19 ± 0.07), lactose (3.96 ± 0.04), protein (2.65 ± 0.02), solids (0.58 ± 0.01), and fat (0.23 ± 0.01) n percent, with a density of 25.7 ± 0,02 kg/m³ and a freezing point of -0.44 ± 0,02 °C”. This information has been added to the supplementary material (figure S1).
Day 1 is more accurate than day 0. Some examples given by the author/authors (day 0), hundreds can be written for day 1.
Answer: Thank you for your consideration. However, we disagree, and as we explained earlier the best to this manuscript is day 0. Based on the literature, we authors should use day 1, if we performed the analyses one day after preparing the products. This was not the case. We conduct the analyses right after the product has been prepared, which is why day 0.
Please review the answers written for the first revision.
Answer: Thank you again. We revised the answers, and new information was added.